# Polycyclic aromatic hydrocarbons (PAHs), oxy- and nitro-PAHs in ambient air of the Arctic town Longyearbyen, Svalbard

Tatiana Drotikova[1,2*], Aasim M. Ali[3], Anne Karine Halse[4], Helena C. Reinardy[5,1], and Roland Kallenborn[2,1]

[1]Department of Arctic Technology, University Centre in Svalbard (UNIS), Longyearbyen, 9171, Norway
[2]Faculty of Chemistry, Biotechnology and Food Sciences, Norwegian University of Life Sciences (NMBU), Ås, 1432, Norway
[3]Department of Contaminants and Biohazards, Institute of Marine Research (IMR), Bergen, 5817, Norway
[4]Department of Environmental Chemistry, Norwegian Institute for Air Research (NILU), Kjeller, 2007, Norway
[5]Scottish Association for Marine Science (SAMS), Oban, Argyll, PA37 1QA, United Kingdom

*Correspondence to*: Tatiana M. Drotikova (tatiana.drotikova@unis.no)

**Abstract.** Polycyclic aromatic hydrocarbons (PAHs) are not declining in Arctic air despite reductions in their global emissions. In Svalbard, the Longyearbyen coal-fired power plant is considered to be one of the major local source of PAHs. Power plant stack emissions and ambient air samples, collected simultaneously 1 km (UNIS) and 6 km (Adventdalen) transect distance, were analyzed (gaseous and particulate phases separately) for 22 nitro-PAHs, 8 oxy-PAHs and 16 parent PAHs by GC/ECNI/MS and GC-MS/MS. Results confirm low level of PAH emissions ($\sum$16 PAHs = 1.5 µg kg$^{-1}$ coal) from the power plant. Phenanthrene, 9,10-anthraquinone, 9-fluorenone, fluorene, fluoranthene, and pyrene accounted for 85 % of the plant emission (not including naphthalene). A dilution effect was observed for the transect ambient air samples, 1.26 ± 0.16 and 0.63 ± 0.14 ng m$^{-3}$ sum all 47 PAH derivatives for UNIS and Adventdalen, respectively. The PAH profile was homogeneous for these recipient stations with phenanthrene and 9-fluorenone being most abundant. Multivariate statistical analysis confirmed coal combustion, vehicle and marine traffic as the predominant sources of PAHs. Secondary atmospheric formation of 9-nitroanthracene and 2+3-nitrofluoranthene was evaluated and concluded. PAHs partitioning between gaseous and particulate phases showed a strong dependence on ambient temperatures and humidity. The present study contributes important data which can be utilized to eliminate uncertainties in model predictions that aim to assess the extent and impacts of Arctic atmospheric contaminants.

## 1 Introduction

Traditionally, Arctic regions are considered to be pristine and remote from the majority of potential large-scale emission sources in industrialized middle latitude countries (Armitage et al., 2011; Macdonal et al., 2000; Barrie et al., 1992). Atmospheric transport is the most efficient way for polycyclic aromatic hydrocarbons (PAHs), released in the lower latitudes, to reach the Arctic (Friedman et al., 2014). Long range atmospheric transport (LRAT) to Arctic regions has strong seasonality with an increased tendency during winter and spring (Willis et al., 2018). This is driven by a different mean circulation direction across the Arctic in winter compared to summer, the extension and  significantly increased permeability of the Arctic front in winter, and the absence of wet removal of particles during transport (Willis et al., 2018). These factors explain observed maximum near-surface pollutant concentrations during winter and minimum levels during summer (Klonecki, 2003). Fossil fuel sources dominate total aerosol organic carbon in Arctic winter air, with a predominance of alkanes, PAHs, and phthalates (Fu et al., 2009). During the past decades, the background monitoring of atmospheric pollutants in Ny-Ålesund, Svalbard, and Alert, Canada, have been an important data repository for information on occurrence and LRAT of anthropogenic contaminants including persistent organic pollutants and PAHs in the Arctic regions. The data demonstrates ubiquitous distribution of PAHs on a global scale, including the Arctic. "Confirmed occurrence of a pollutant in a polar environment" is an important criterion considered by conventions, including the United Nations

Economic Commission for Europe (UNECE), Stockholm, Basel, and Rotterdam (Fiedler et al., 2019). UNECE has incorporated PAHs in the Convention on Long-range Transboundary Air Pollution (UNECE, 1998). Atmospheric PAHs are regulated in USA, Canada, UK, and Europe (US EPA, 2011; Ontario Ministry of the Environment and Climate Change, 2016; UK Air DEFRA, 2007; EU Directive 2004/107/EC, 2005). PAHs are also included in the list of target chemicals of the Convention for the Protection of the Marine Environment of the North-East Atlantic (OSPAR). PAH concentrations are not declining in the Arctic despite global emission reductions (Yu et al., 2019), and PAHs are listed as "chemicals of emerging concern in the Arctic" (Balmer and Muir, 2017).

PAHs are byproducts of different incomplete combustion processes, mainly fossil fuels and biomass burning (Ravindra et al., 2008a). Their toxic and carcinogenic effects on both human health and ecosystems are well documented (Kim et al., 2013; Reynaud and Deschaux, 2006; Macdonald et al., 2010). Under unique Arctic weather conditions, with extreme temperatures, wind, and light seasonality, atmospheric PAHs may behave differently compared to in temperate climatic conditions. Low temperatures favor partitioning of semi-volatile PAHs from gas phase to particulate phase, which makes them more persistent in the Arctic environment (Lammel, 2015). Due to extended winter darkness in the Arctic, photodegradation of PAHs is limited for several months. The transition from dark polar winter to the light spring and summer brings large increase in the amount of available solar radiation and oxidants in the Arctic troposphere (Willis et al., 2018). PAHs react with a number of atmospheric oxidants, most notably the hydroxyl radical, ozone, the nitrate radical, and nitrogen dioxide (Keyte et al., 2013). This leads to their transformation into more toxic oxygenated and nitrated PAH derivatives (oxy-PAHs and nitro-PAHs). Oxy- and nitro-PAHs are also constituents of raw coal and can be emitted with PAHs following the same combustion processes (Huang et al., 2014b). Oxy- and nitro-PAHs have high toxicity (Onduka et al., 2012); they can act as direct mutagens, carcinogens, and oxidative stressors in biota (Durant et al., 1996). The biological effects of nitro- and oxy-PAHs can be greater than those of the parent PAHs (WHO, 2003). In remote locations they are found at concentrations near detection limits and thus are mostly not included in monitoring programs, and the level of nitro- and oxy-PAHs in the Arctic atmosphere is unknown (Balmer and Muir, 2017).

The Arctic is warming at a higher rate than the global average and visible changes happen rapidly here. Thus, it is a key area for modeling studies on climate effects on contaminants with a main focus on LRAT from lower latitudes. As a consequence, local Arctic sources are usually disregarded, and lack of information on local emission sources is a source of uncertainty in model predictions that often deviate significantly from observations (Schmale et al., 2018). Local emission sources may be of high importance in winter, when strong temperature atmospheric inversions can be frequent in Arctic region (Bradley et al., 1992). These episodes inhibit the mass and heat fluxes from the surface to the atmosphere, and consequently the dilution of surface emissions (Janhall et al., 2006; Li et al., 2019). This trapping of emissions results in poor air quality and can be potentially harmful to local people. Climate change introduces additional sources of PAHs to the Arctic region. In the past decade, human activities such as resource exploration, research, tourism, fisheries, and maritime traffic have increased substantially due to warming and corresponding reduction of sea ice, opening up new shipping routes (Jörundsdóttir et al., 2014). Warming may also enhance volatilization of low molecular weight (LMW) PAHs from ground surfaces (Friedman et al., 2014) and melting sea ice (Yu et al., 2019). Reactivity of PAHs in the gas phase is significantly greater than when associated with particles (Keyte et al., 2013), therefore increasing air temperatures can be expected to lead to increased levels of toxic nitro- and oxy-PAHs.

The need for a comprehensive assessment of local contaminant sources in Svalbard was acknowledged and initiated in the international Arctic Monitoring and Assessment Programme (AMAP); with the major focus on persistent organic pollutants (Pedersen et al., 2011), there is a scarcity of data on local sources of PAHs in Svalbard. A back-trajectory analysis of twenty years data for thee representative PAHs (Phe, Pyr, and BaPyr; see Table 1 for full names) suggested that Svalbard is impacted by air masses coming from eastern Russia, northern Europe, and northwest Russia during winter (Yu et al., 2019). Overall, combined European and Russian emissions accounted for more than 80 % of episodic high-concentration events in

Svalbard in 2007 (Balmer and Muir, 2017; Friedman and Selin, 2012). However, observed concentrations of Phe and Pyr from the Zeppelin station, Svalbard, were higher than model simulations, indicating important contributions of local sources of PAHs to the Arctic atmosphere, too (Yu et al. (2019). This study focused on the main settlement in Svalbard, Longyearbyen, with a population of approximately 2400 inhabitants and a high level of (partially seasonal) human activities (transport, coal mining, industry, tourism, and research). The local coal-fired power plant (PP) was hypothesized to be the major local source of PAHs, and the overall objectives of this study were to: (1) evaluate PAH emissions from the local power plant, (2) examine concentrations and profile changes with distance from the PP, (3) quantify concentrations of PAHs and nitro- and oxy-PAHs, in both gaseous and particulate phases, and (4) determine other potential local sources of PAHs, and nitro- and oxy-PAHs.

## 2 Material and methods

### 2.1 Sampling site

Svalbard is an archipelago located between latitudes 77° and 81°N in the Western Barents Sea. Longyearbyen, being the largest populated settlement, was chosen as the study area. The local PP was installed in Longyearbyen in the 1980s and provides the community with sufficient electricity (45 000 MW) and central heating supply (70 000 MW) throughout the year (Bøckman, 2019). The PP is fueled by coal produced in a nearby mine at Breinosa (mine No. 7). This coal has a distinct quality (brown, high volatile bituminous coal with vitrinite reflectance $R_o$=0.78 % (Marshall et al., 2015)) and is well suited for energy production. Coal consumption is about 25-30 thousand ton per year. The PP has two boilers, 32 MW each. The coal burning temperature is about 1000 °C (Bøckman, 2019). Since December 2015, the flue gas purification system consists of a selective non-catalytic reduction (SNCR) system, an electrostatic precipitator (ESP), and a wet flue-gas desulfurization (WFGD) scrubber. After SNCR the $NO_x$ content in the flue gas is reduced by 50 % by spraying urea solution as a reduction agent into the boiler. Further, in the ESP step, dust is electrically charged and deflected toward the collection electrodes. In the WFGD scrubber, the flue gas is cooled and desulfurized by sea water. Low emissions are reported: dust 1.5±0.2, $SO_2$ 0.3±0.1, $NO_x$ 244±19, CO 63±5 mg $Nm^{-3}$ (Lundgjerdingen, 2017), reflecting high efficiency of the flue gas cleaning system. For PP emission analysis, stack emission air samples were collected at source (PP), and two locations at transect distance: the roof of the University Centre in Svalbard (UNIS, urban location, 1 km from PP) and the former northern lights observatory in Adventdalen (Adventdalen, rural location, 6 km from PP, 7 km to the active coal mine No. 7; Fig. 1). Sampling at UNIS and Adventdalen was performed simultaneously.

### 2.2 Sample collection

#### 2.2.1 Power plant

A total of 6 low volume (1.3-3.0 $m^3$) samples of the PP stack emission were collected (Table S1) under normal operating conditions, collected on 27[th] September (PP1-PP3) and 2[nd] October (PP4-PP6) 2018. Sampling was performed downstream the WFGD scrubber, after all flue gas cleaning steps. The sampling probe (inner Ø = 11 mm) was situated to face the direction of the flue gas. A custom-made low volume, battery powered, air sampler (Digitel, Switzerland) was used to pump the flue gas through the sampling material placed in a stainless steel cartridge (16249, Sartorius Stedim Biotech GmbH, Germany). The particulate phase was collected on quartz fiber filter (QFF, pre-burned at 450°C for 6 h; Ø = 47 mm, no binder, Pallflex, USA), and the gaseous phase on polyurethane foam (PUF, Soxhlet pre-cleaned in toluene for 24 h followed by 24 h acetone wash; Ø = 50 mm, L = 75 mm, Klaus Ziemer GmbH, Germany). Although, the pump was operated at the maximum speed (35 L $min^{-1}$, which corresponds to 6.1 m $s^{-1}$ probe intake flow speed), an isokinetic sampling regime was not achieved. The flue gas parameters (temperature 8.9±0.5°C, moisture 28±2 %, flow speed 18.1±0.8 m $s^{-1}$, and density

$1.24\pm0.2$ kg m$^{-3}$) were measured during the sampling by FKT3DP1A multi meter equipped with S-type Pitote probe (FlowKinetics LLC, USA).

**2.2.2 UNIS and Adventdalen**

The prevailing wind direction in Longyearbyen and Adventdalen is from the southeast. In summer, when the soil surface in Adventdalen becomes warmer than water surface in Adventfjørd, the wind direction can temporary change to northwesterly (Dekhtyareva et al., 2016; Esau et al., 2012). To focus on PP emission and avoid the peak of marine traffic in the summer, simultaneous sampling at UNIS and Adventdalen was carried out from 28[th] August to 28[th] of September 2018 (Table S1), on days with predicted northwesterly wind direction (Fig. S1), using high volume air samplers (TISCH-1000-BLXZ, TISCH Environmental Inc., USA) equipped with dual chamber sampling module (particle filter, stainless screen and vapor filter, glass cartridge). About 370 m$^3$ of ambient air was collected over 24 h per sample (Table S1). For each station, 6 high volume air samples were collected for particulate (QFF, pre-burned at 450°C for 6 h; Ø = 103 mm, no binder, Munktell/Ahlstrom, Finland) and gaseous (PUF, Soxhlet pre-cleaned in toluene for 24 h followed by 24 h acetone wash; Ø = 65 mm, L = 100 mm, Klaus Ziemer GmbH, Germany) phases. Weather parameters including ambient temperature, relative humidity, UV radiation, wind direction, and precipitation were recorded (Table S2). All samples (PP, UNIS, and Adventdalen) were kept intact inside the sampling unit after collection. In order to reduce the risk of post-collection contamination, the unit was sealed in two plastic bags for transportation to the lab, where samples were removed from the unit, sealed with layers of aluminum foil, and stored airtight in two plastic bags. Samples were kept frozen at -20°C until analysis. A total of 18 samples (18 QFFs and 18 PUFs) and 8 field blanks (4 for PP and 4 for UNIS and Adventdalen) were collected.

**2.3 Analytical procedure**

16 PAHs, 8 oxy- and 22 nitro-PAHs (Table S3) were quantified using gas chromatography in combination with electron ionization mass spectrometry GC-EI-MS/MS and gas chromatography in combination with electron capture negative ion mass spectrometry GC-ECNI-MS, respectively. Full details on analytical methods, including equipment and procedures, are outlined in the SI (Text S1, Tables S4 and S5). In brief, all QFFs (particulate phase) and PUFs (gaseous phase) samples were extracted separately by two different methods, followed by the same clean-up procedure. Several $^2$H-labelled PAH (dPAH) surrogates (16 dPAHs, 3 dOxy-PAHs, and 6 dNitro-PAHs), were added to samples prior to extraction. QFF samples were extracted with dichloromethane using a quick easy cheap effective rugged and safe QuEChERS-like procedure developed previously for the analysis of particulate bound PAHs (Albinet et al., 2013; Albinet et al., 2014). PUF samples were Soxhlet extracted with dichloromethane for 24 h. The extracts were concentrated and cleaned up first with neutral alumina Al$_2$O$_3$, and then with neutral silica SiO$_2$. Elutes were dried under gentle nitrogen stream and redissolved in approximately 100 μL $n$-hexane. The purified samples were spiked with three labelled standards to evaluate the surrogate recoveries.

**2.4 Quality assurance**

Detailed information on method validation and quality control is provided in SI (Text S2). Field (n=4 for PP, n=4 for UNIS and Adventdalen combined) and laboratory (n=3 for PP, n=3 for UNIS and Adventdalen combined) blanks were analysed in order to evaluate possible contamination during sample transport and analysis. The method detection limit (MDL) was determined based on blank values for each sampling material type (Table S6). High contamination of PUF blank samples by Nap and 9,10-PheQ for UNIS and Adventdalen was found; these compounds were excluded from the final results. No blank correction was performed for the concentration calculations. Samples with PAH concentrations below instrumental limit of quantification (LOQ) were replaced by LOQ/2 for statistical analysis. The method efficiency was tested using QFF (n=4) and PUF (n=4) spiked samples (Table S7). Acceptable recoveries ranged between 63-109 % for dPAHs, 56-68 % for dOxy-PAHs, and 44-89 % for dNitro-PAHs (Table S8).

## 2.5 Statistical analysis

Statistical analyses of compound concentrations were performed with Minitab 18 Statistical Software (Minitab LLC, Pennsylvania, USA). Normality and homogeneity of variances were tested with Shapiro-Wilk and Levene's tests, respectively. Mann–Whitney U test was performed to test significant differences between sampling locations. Spearman's correlation was used to investigate relationships between different variables. The statistical significance was set at $p < 0.05$, unless stated.

Principle component analysis (PCA) was performed for PAH source apportionment. A $6 \times 29$ matrix (sample number $\times$ 29 detected compounds, including 14 PAHs and 15 nitro- and oxy-PAHs) dataset was used to assess the source contribution to PAHs for each location (UNIS and Adventdalen). Total PAH concentrations (gaseous and particulate, G+P), were used to minimize the influence of partitioning, ageing, and photochemical degradation (Kim et al., 2009). PCA was based on a correlation matrix to standardize scales and weight all variables equally (Holmes et al., 2017). PCA was first applied on the concentrations matrix only, and then additional parameters (weather and diagnostic ratios) were carefully included in order to explain the observed sample groupings.

## 3 Results and discussion

### 3.1 Longyearbyen power plant PAH emission profile

Individual concentrations and phase distribution (percentage on particulate matter, %PM) of target PAHs are summarized in Table 1. The sum of total (G+P) concentration of the 16 priority PAHs ($\sum$16 PAHs; U.S. Environmental Protection Agency) in the purified flue gases emitted from the PP is 0.106 µg m$^{-3}$, which corresponds to 1.5 µg kg$^{-1}$ coal. Currently, there is no PAH emissions standard for coal-fired power plants in Norway. However, compared to the Canadian emission limits of PAHs for municipal solid waste incinerators of 5 µg m$^{-3}$ (Li et al., 2016), the Longyearbyen PP emissions is a factor of 3 lower. About 94 % of 16 PAHs were emitted in a gas phase, in agreement with earlier studies (Li et al., 2016; Wang et al., 2015; Yang et al., 1998). The emission profile of the Longyearbyen PP is dominated by LMW PAHs (2 and 3 rings), which represents 89 % of $\sum$16 PAHs emission; high molecular weight (HMW) PAHs (5-7 rings) were not detected, likely due to their low vapor pressure and thus association to particles. A combination of ESP and WFGD has a removal efficiency of PM up to 99.9 % (Wang et al., 2019). Fine cooling of the PP flue gas ($8.9 \pm 0.5$ °C) by cold sea water facilitates high PM collection efficiency as well (Noda and Makino, 2010; Wang et al., 2019). As a result, PP dust emissions are below the ultra-low standard of 5 mg m$^{-3}$ (Zhao et al., 2017) at $1.5 \pm 0.2$ mg m$^{-3}$ (Lundgjerdingen, 2017). The PAH emissions profile was dominated by Nap and Phe, accounting for 53 % and 27 % of $\sum$16 PAHs, followed by Flu, Flt and Pyr. Nap and Phe are often reported as major emitted compounds from power plants equipped with analogous exhaust cleaning systems and/or burning the same type of coal (Hsu et al., 2016; Li et al., 2016; Wang et al., 2015). A similar PAH emissions profile was reported by Hsu et al. (2016) for the power plant in central Taiwan (Table S12). A higher flue gas dust concentration and different coal sources resulted in 40 % emissions of four ringed PAHs compared to 11 % for Longyearbyen PP. Operation conditions and boiler type can have significant effects on emitted PAH profiles and concentrations (Wang et al., 2015), as well as combustion temperature (Peng et al., 2016), and geological maturity (Huang et al., 2014b).

Nitro- and oxy-PAHs are constituents of raw coal and can also be produced from parent PAH compounds during high temperature coal combustion (Huang et al., 2014b). The yields of individual nitro-PAHs from the PP was 1 - 2 orders of magnitude lower than those of their corresponding parent PAHs, and individual concentrations were at or below 1.7 ng m$^{-3}$; 1-NNap was the most abundant nitro-PAH. Huang et al. (2014b) investigated the same type of coal (bituminous, $R_0 = 0.77$ %), burned at lower temperatures in a honeycomb briquette stove; nitro-PAHs were absent in the raw coal and calculated nitro-PAH/PAH ratios were >1 confirming formation of nitro-PAH compounds during coal combustion. In contrast, in the

present work, the same daughter to parent PAH ratios were < 1 (Table 2), indicating an absence of nitro-PAH formation during coal combustion or possible thermal degradation of nitro-PAH at 1000 °C.

The yields of oxy-PAHs were orders of magnitude higher than nitro-PAHs because oxy-PAHs can be produced by reaction of PAH with $O^·$ or $^·OH$ radicals generated continuously by radical chain reactions during combustion (Huang et al., 2014b). 9-Flu and 9,10-AntQ were the most abundant among the oxy-PAHs (12.4 and 15.6 ng m$^{-3}$, respectively), and concentration of 9,10-PheQ was a factor of six lower. The calculated ratios of oxy-PAH to corresponding parent PAH were lowest for 9,10-PheQ/Phe and highest for 9,10-AntQ/Ant (Table 2). This can be due to a higher content of Phe in coal, as well as

different reaction rates of Phe, Ant, and Flu with $O^·$ or $^·OH$ radicals. Difference between the reaction rates of Flu and Ant can possibly be explained by different reaction pathways;  Flu undergoes H atom abstraction at the 9-position to form 9-Flu, while Ant requires $^·OH$ attack on the aromatic ring (Brubaker and Hites, 1998). Ant and Phe have essentially the same 3-ring structure, only differing by the relative position of their aromatic rings. However, Ant appears to be significantly more reactive,  due to the sterically unhindered molecular structure of Ant (Keyte et al., 2013). Formation of specific PAHs is also

a temperature dependent process (Peng et al., 2016).

Ant, BaAnt, and Chry are often used as indicators of coal combustions (Zheng et al., 2019; Wu et al., 2014; Wang et al., 2009), however, their concentrations in the flue gas of the Longyearbyen PP were negligible. This demonstrates the strong importance of determining indicatory PAH profiles for individual combustion sources for correct source identification. PAH emissions from different coal plants are hard to compare because they are affected by many factors including coal type,

boiler load, combustion mode (Wang et al., 2015), and flue gas cleaning systems. Nap was the most abundant PAH emitted from the Longyearbyen PP. Due to its ubiquitous presence, Nap was not considered as suitable marker. Further, Phe, Flu, Flt, Pyr, 9-Flu, and 9,10-AntQ were the main PAHs and oxy-PAHs detected in the Longyearbyen PP flue gas (Fig. 2), therefore the  presence and  diagnostic ratios (Table 3) of these compounds were used as markers of the PP source in the present work. In Yu et al. (2019), coal combustion was identified as the main source (68 % contribution) of PAHs, at the Zeppelin

monitoring station at Ny-Ålesund, Svalbard, and Phe, Flu, Flt, and Pyr were the main contributors, most likely attributable to the Longyearbyen PP located 115 km southeast of Ny-Ålesund. Overall, total flue gas emissions were 960 000 Nm$^3$ day$^{-1}$ (Lundgjerdingen, 2017), and a daily emissions of Σ16 PAHs and sum of the nitro- and oxy-PAHs are approximately 98.7 g of and 35.6 g, respectively.

### 3.2 UNIS and Adventdalen

**3.2.1 Ambient concentrations and PAH profiles**

The concentrations of PAHs and oxy-PAHs measured at UNIS were a factor of 2 higher than at Adventdalen, while nitro-PAH levels differed less (Table 1). Σ15 PAHs were 749.2±72.6 (UNIS) and 369.1±66.7 pg m$^{-3}$ (Adventdalen); Σ7 Oxy-PAHs were 471.0±150.8 (UNIS) and 233.1±68.3 pg m$^{-3}$ (Adventdalen); Σ22 nitro-PAHs were an order of magnitude lower than both parent PAHs and oxy-PAHs, with average values of 36.8±6.2 (UNIS) and 27.2±11.1 pg m$^{-3}$ (Adventdalen). The

UNIS and Adventdalen chemical profiles of PAHs and oxy-PAHs were similar, while the profiles of nitro-PAHs differ (Fig. S2). Proportions of 1- and 2-NNap were higher in UNIS samples (about 60 % of Σ22 nitro-PAHs), while 9-NAnt and 2+3-NFlt showed higher contributions into the nitro-PAH profile of samples from Adventdalen (about 55 % of Σ22 nitro-PAHs). Among the parent PAHs, Phe (ranging from 191.7 to 470.0 pg m$^{-3}$) and Flu (ranging from 38.5 to 236.0 pg m$^{-3}$) were the most abundant in the present study. The Phe and Flu concentrations measured in Longyearbyen (UNIS and Adventdalen)

were 2 orders of magnitude higher than those detected at the Zeppelin station and the same order of magnitude as in Birkenes (southern mainland Norway) for the same period  (Table S13). The PAH profiles were dominated by Phe and Flu at all sites (Fig. S3). A higher proportion of Phe was observed in Longyearbyen samples. The measured PAH concentrations in the present study were in agreement with the 2 decades average data reported for the Arctic monitoring stations in Svalbard (Zeppelin) and Finland (Pallas), and were about an order of magnitude higher compared to the Canadian Arctic (Alert)

concentrations (Yu et al., 2019). The PAH levels observed in Longyearbyen were significantly (up to 2 orders of magnitude) lower compared to rural sites in Europe and China (Table S14).

Among measured oxy-PAHs, concentrations of 9-Flu and 9,10-AntQ were the highest in the present study. The 9-Flu level ($270.3\pm146.9$ pg m$^{-3}$ at UNIS and $139.4\pm24.9$ pg m$^{-3}$ in Adventdalen) was a factor of 3 higher than reported for Pallas and Råö (southern Sweden) background stations (Brorström-Lundén et al., 2010), while 9,10-AntQ ($163.5\pm57.4$ pg m$^{-3}$ at UNIS and $71.7\pm39.2$ pg m$^{-3}$ in Adventdalen) in Longyearbyen was equal to the winter levels in central European background air (Lammel et al., 2020). The sum of oxy-PAHs detected at UNIS was similar to rural sites in eastern England (Alam et al., 2014) and the central Czech Republic (Lammel et al., 2020), but were significantly lower than in rural southern China (Huang et al., 2014a) and the French Alps (Albinet et al., 2008).

1-NNap and 2+3-NFlt were the most abundant nitro-PAHs detected at UNIS and Adventdalen. The level of 2+3-NFlt ($9.5\pm1.6$ pg m$^{-3}$ at UNIS and $12.3\pm7.7$ pg m$^{-3}$ in Adventdalen) was an order of magnitude higher than that at Råö and Pallas stations (Brorström-Lundén et al., 2010), and 1-NNap average concentrations were $17.0\pm3.0$ pg m$^{-3}$ at UNIS and $5.0\pm3.2$ pg m$^{-3}$ in Adventdalen. Overall, nitro-PAH concentrations were similar to those reported for the Pallas and Råö Scandinavian stations (Brorström-Lundén et al., 2010), and the rural site in the Czech Republic which is representative of central European background levels (Lammel et al., 2020).

**3.2.2 Gas/particle partitioning**

Gas/particle partitioning is an important process that controls transport, degradation, and distribution patterns of contaminants in and between environmental compartments (Finlayson-Pitts and Pitts Jr, 1999; Lammel et al., 2009; Franklin, 2000). The sampling campaign in the present study was conducted from late Arctic summer until early autumn and during this period the air temperature varied from +6.8 °C in August to -4.4°C in September and several precipitation events (snow and rain) occurred. In general, LMW PAHs were found in the gas phase, while HMW PAHs were present in the particulate phase (Table 1), which is in accordance to their physico-chemical parameters, such as octanol-air partition coefficient, vapor pressure, and molecular weight (Table S3; Tomaz et al., 2016; Shahpoury, 2016). Repartitioning between phases (Fig. 3) mainly impacted semi-volatile compounds with three and four aromatic rings (Flt, Pyr, BaAnt, Chry; 2-NFlu, 9-Flu, cPPhe-4, 9,10-AntQ, 9-NAnt, and 2+3-NFlt) as a response to changing meteorological conditions (Hu et al., 2019). Strong negative correlations of percentage of PAH determined in particulate phase (%PM) with ambient temperature were confirmed for most of these compounds (Table S15). %PM also depends on aerosol surface area, organic matter, and black carbon content (Lohmann and Lammel, 2004).

Compared to Adventdalen, the urban UNIS location ensure a higher level of PAHs emitted from different nearby anthropogenic sources, including the PP. Furthermore, low ambient temperature reinforces partitioning of freshly emitted gaseous PAHs to the particulate phase. As a result, %PM at UNIS was higher than in Adventdalen. Deposition (wet and dry) and chemical reactions with atmospheric oxidants are important removal processes of PAH from air (Keyte et al., 2013). On the local scale, within an hour of travel time from PP to Adventdalen, it is not expected that photolytically-initiated transformation of the freshly emitted PAHs has a strong influence on gas phase concentrations and consequently on %PM. Dry deposition rates vary depending on the type of adsorbing particle (mass, size, aerodynamic properties, shape, and chemical composition) and the atmospheric conditions (Weinbruch et al., 2018), and may be a dominant PAH removal process in source areas (Sharma and McBean, 2002).

The influence of wet deposition was indicated by a significant negative correlation between amount of precipitation and concentrations of several particle-bound HMW PAHs (Chry, BbkFlt, IPyr, BPer, BaFlu-11, and BaAnt-7,12) as well as semi-volatile Phe, Flt, and Pyr, which are more predominant in gaseous phase (Spearman correlation, $p<0.05$, Table S16. Effective wet scavenging of Phe, Flt, and Pyr has been earlier suggested (Škrdlíková et al., 2011). Furthermore, a strong negative correlation with mass of water vapor in the air (specific humidity) was determined for most of the compounds

(Spearman correlation, p<0.05, Table S17). Particle associated HMW compounds are readily scavenged by precipitation, while water solubility and polarity (for nitro- and oxy-PAHs) play an additional role in wet scavenging processes (Shahpoury et al., 2018). The gas phase removal from the atmosphere is due to substance dissolution in water droplets, which enhances the scavenging effect at higher humidity. Higher sensitivity of gas scavenging compared with particle scavenging towards liquid water content was also indicated by Škrdlíková et al. (2011).

In general, the obtained %PM were in agreement with the earlier reported (Table S18). Higher %PM of 9,10-AntQ and several nitro-PAHs (1- and 2-NNap, 2-NFlu, 9-NAnt, 9-NPhen) were detected in French Alpine sites in winter (Albinet et al., 2008), while higher %PM of Flt, 9-Flu, and cPPhen-4 found in the present study contrasts with those reported for temperate urban and rural sites in China and Europe (Huang et al., 2014a; Tomaz et al., 2016). Sources difference, weather influence such as precipitation and temperature, as well as different atmospheric conditions (e.g., number of suspended particles, mass size particle distribution, and specific humidity), are likely responsible for these variations.

## 4 Source identification

Due to changes in the Arctic front, more frequent precipitation, and low levels of wood and coal burning for residential heating in the northern hemisphere in the summer, the LRAT of PAHs to the Arctic is low in summer. Sampling was performed on days with predicted northwesterly wind, and according to the 5-day back trajectory analysis, the air arriving to Longyearbyen in the sampling period mainly came from the north and from Greenland (Fig. S4). As discussed in Section 3.2.1, up to two orders of magnitude lower PAH concentrations were detected at the Zeppelin monitoring station compared to the levels in Longyearbyen on the same time. Thus, local emissions were the main sources of PAHs in Longyearbyen in this study.

Besides the PP emission, vehicles are another obvious local source of PAHs. In 2018, 1558 vehicles, including cars, lorries, and busses, were registered in Longyearbyen (Table S19; Statistics Norway, 2018). Longyearbyen maintains about 50 km of paved and unpaved roads dedicated for traffic (Bore R.R., 2012). Sampling was conducted at the end of summer in order to avoid peak emissions from marine traffic and to focus on PP emissions; however, it is likely that some of the 718 registered private boats were active in Adventfjørd and several larger ocean-going vessels were in the port around the sampling period (weeks 34-38, Fig. S5). Thus, shipping emissions could not be eliminated as a potential source of PAHs. To note, there is no local waste incineration and wood burning.

PCA was applied to samples from Adventdalen (n=6) and UNIS (n=6) to determine potential PAH sources at each location. Total PAH (G+P) concentrations were used to minimize the influence of partitioning, aging, and photochemical degradation. Selected PAH diagnostic ratios (Table 3) and weather parameters were utilized as additional supportive tools for sources interpretation, and their values were used as variables. Diagnostic ratios may be affected by large-scale mixing of PAHs in the atmosphere, differing emission rates of PAH from the same source, influence of changing environmental conditions, and atmospheric processing of individual PAH compounds with different atmospheric lifetimes and reactivities (Alam et al., 2013; Tobiszewski and Namieśnik, 2012; Katsoyiannis and Breivik, 2014). Ratios based on highly reactive compounds such as Ant and BaAnt were not included, while more stable HMW PAHs diagnostic ratios were interpreted with greater confidence (Galarneau, 2008; Alam et al., 2014). Yunker et al. (2002) previously proposed the ratio of IPyr/(IPyr+BPer) to recognize vehicle from coal combustion emissions. BbkFlt/BPer was selected as an additional marker ratio for traffic due to the greater capacity to discriminate diesel and gasoline emissions, as well as its wider value range (Kuo et al., 2013). The Flt/(Flt+Pyr) ratio is often used for source identification and, in particular, to understand if PAHs are mainly emitted from petroleum sources or from combustion processes (Yu et al., 2019). The Flu/(Flu+Pyr) ratio was selected as a specific indicator for coal combustion due to its strong correlation with the local PP determined markers, and the ratio value was also

in agreement with literature (Yunker et al., 2002; Katsoyiannis and Breivik, 2014). Two principal components (PCs) for Adventdalen (74 %) and two PCs for UNIS (74 %) were focused on.

**4.1 Adventdalen**

The first and the second PCs described 51 % and 23 % of the total variance, respectively (Table S20). Three groups of compounds suggest three different potential sources. The first group include Flt, Pyr, cPPhe-4, BaFlu-11, and BaAnt-7,12. Strong correlations between their concentrations and the IPyr/(IPyr+BPer) ratio suggest a traffic origin for these compounds (Yunker et al., 2002), and specifically diesel emissions (Table 3; Table S21; Ravindra et al., 2006; Ravindra et al., 2008a). Because of the rural position, car traffic is much lower at this location. At the same time, due to the proximity to an active

mine (Fig. 1), heavy-duty vehicles (coal trucks, tourist busses, geotechnical drilling machinery) are thus the main candidate source for PAH vehicle emissions. Produced coal is regularly delivered from the mine to PP and storage area in the harbor on a road situated in 150 m distance from the Adventdalen sampling station. Coal is transported by Volvo FH540 trucks (built in 2018-2020) driven on diesel CFPP-12 (NS-EN 590; Nilssen P., Store Norske, personal communication; Table S23). The trucks have Euro 6 standard compliant Volvo D13K engines (HC 0.13, CO 1.5, $NO_x$ 0.4, PM 0.01 g $(kWh)^{-1}$; DieselNet,

2020) fitted with exhaust gas recirculation, diesel particulate filter, diesel oxidation catalyst, selective catalytic reduction, and ammonia oxidation catalyst (Volvo Trucks, 2020). These allow high operation temperatures and high efficiencies in reducing particle and $NO_x$ emissions. Numerous studies showed substantial reduction in gaseous and particulate emissions of PAH, nitro- and oxy-PAHs as the result of such mitigation in particle and $NO_x$ emissions (Hu et al., 2013; Gerald Liu et al., 2010; Khalek et al., 2015; Huang et al., 2015). Up to 10 orders of magnitude reduction in emission from similar to Volvo

D13K heavy-duty engine was reported for several nitro-PAHs (6-NChry, 1-NPyr, 2-NPyr, 4-NPyr, 7-NBaAnt; Liu et al., 2015; Gerald Liu et al., 2010), which were not detected in the present study most likely due to low vehicle number in Adventdalen. However, Flt, Phe, and Pyr have been widely reported to be emitted after diesel emissions (Albinet et al., 2007; Ravindra et al., 2008a; Wingfors, 2001) and BaFlu-11, BaAnt-7,12, cPPhen as well (Nyström et al., 2016; Ahmed et al., 2018; Rogge et al., 1993). 9-Flu, 9,10-AntQ, and 1-NNap were the main oxy- and nitro-PAHs emitted from modern

technology heavy-duty diesel engine (Liu et al., 2015; Gerald Liu et al., 2010; Guan et al., 2017), supporting the traffic origin of the group 1 compounds too.

9-Flu, 9,10-AntQ, and 1-NNap, together with Phe and Flu (group 2), seems to have a double origin. On a PCA loading plot these compounds have similar proximity to the traffic emission ratio (PC1), as well as to the coal combustion ratio (PC2). As earlier reported, Phe, Flu, 9,10-AntQ, 9-Flu, and 1-NNap together accounted for 74 % of the total PAH emission from the

local PP (Fig. 2) and Mann-Whitney U statistical test (n=6, p≤0.05) showed no significant difference between values of the Flu/(Flu+Pyr) coal combustion diagnostic ratio based on the measured Flu and Pyr concentrations in Adventdalen and in the local PP stack emission.

The PC1 emphasized a positive correlation of 2+3-NFlt and 9-NAnt (group 3) with temperature, humidity, and UV radiation, as well as a negative correlation with primary PAHs (Fig. 4, Table S17), suggesting a secondary source of origin. The

daughter-to-parent PAH ratios, 9-NAnt/Ant and 2+3-NFlt/Flt (Table S24), showed statistically significant correlations with temperature, humidity, and UV radiation (Spearman correlation, p<0.10; Table S25). Moreover, 2+3-NFlt and 9-NAnt had a strong positive correlation with each other and negatively correlated with their parent compounds (Spearman correlation, Table S26), by reason of assumed chemical transformation. It should be noted that 9-NAnt and 2+3-NFlt were detected in the PP flue gas at low levels (0.08 ng $m^{-3}$ and 0.5 ng $m^{-3}$, respectively), and further statistical analysis (Spearman correlation,

Table S26, Fig. 4) showed no correlation with established PP tracers (Phe, 9,10-AntQ, and 9-Flu), suggesting a different source of origin. These results indicate atmospheric formation as an additional source of 9-NAnt and 2+3-NFlt, in agreement with other studies (Lin et al., 2015; Hayakawa et al., 2000; Shahpoury et al., 2018). Sampling close to a major source of $NO_x$ emission such as the local power plant can result in concentrations of $NO_3$ and $NO_2$ at high enough levels for atmospheric

transformation of PAHs to occur (Keyte et al., 2013). Relative contribution of primary and secondary sources of nitro-PAHs
could be evaluated by applying a 2-NFlt/1-NPyr ratio (Ringuet et al., 2012; Ciccioli et al., 1996), but 1-NPyr was not detected in our study.

The two PCs explain 74 % of the total variance. Traffic emission (mainly diesel exhaust) and the coal-burning PP are concluded as the main local sources of PAHs and nitro- and oxy-PAHs in Adventdalen, and atmospheric transformation of PAHs is an additional source of nitro-PAHs.

**4.2 UNIS**

The proximity of the UNIS sampling location to central Longyearbyen, as well as to the PP and the port, makes the UNIS location more complex for source identification. This site is mainly influenced by passenger car traffic, although heavy-duty vehicles also pass UNIS. 1114 private cars were registered in Longyearbyen in 2018 (Statistics Norway, 2018), including old and modern (Euro 3-7 emission standard) technology cars, approximately equally balanced between gasoline and diesel fuel. Gasoline 95 (with up to 5 % bioethanol) and diesel CFPP-12 (with up to 7 % biodiesel) are the exclusive fuels used in summer time, and comply with the Norwegian standard NS-EN 228 and NS-EN 590, respectively, with ultra-low (< 10 ppm or 0.001 %) sulfur content (Storø J., LNS Spitsbergen, personal communication). Details of the fuel parameters can be found in Tables S22 and S23.

Muñoz et al. (2018) undertook a study under similar vehicle and fuel conditions to Longyearbyen and reported a predominance of LMW PAHs for both fuels. Flt, Pyr, Phe and BPer, BaPyr, Chry, BbFlt were found to be the most abundant compounds in gaseous and particulate phases respectively, in agreement with earlier studies (Nyström et al., 2016). Similar PAHs pattern was found for UNIS samples. On the PCA plot (Fig. 5) Flt, Pyr, BPer, BaPyr, Chry, and BbkFlt are grouped together (group 1) and have equally high loadings on the PC1 (Table S27). The compound concentrations are significantly correlated with the traffic ratio BbkFlt/BPer ($p < 0.05$, Table S28) suggesting the same origin of the compounds. The BbkFlt/BPer ratio varied from 0.43 to 0.72 indicating either diesel or gasoline emissions (Kuo et al., 2013). The diesel emission predominance was found for two out of the six sampling days, although particulate phase 1-NPyr, a marker of diesel emissions, was not detected. 1-NPyr forms in the combustion chamber of diesel engines by the addition of nitrogen oxide or nitrogen dioxide to free Pyr radicals (IARC, 2014). Its generation is facilitated by the high engine temperatures (IARC, 2014; Karavalakis et al., 2010; Guan et al., 2017; Huang et al., 2015), which likely can not be reached in Longyearbyen due to short driving distances and low speed limit. The use of high quality ultra-low sulfur fuel with substantially reduced emissions of $NO_x$ leads to reduced nitration of PAHs during fuel combustion (Heeb et al., 2008; Zhao et al., 2020b), and together with low total vehicle number, resulting in low nitro-PAH emissions. Occurred atmospheric deposition may be of influence too.

Gaseous phase 1-NNap and 2-NNap have large loadings on the PC1. They are often reported in traffic emissions (Alam et al., 2015; Albinet et al., 2007; Keyte et al., 2016), as well as oxy-PAHs such as BaFlu-11, BaAnt-7,12 and BZT (Nyström et al., 2016; Albinet et al., 2007; Ahmed et al., 2018; Karavalakis et al., 2010). All these nitro- and oxy-PAHs have a strong positive correlation with the traffic ratio (Table S28). Thus, we conclude traffic (diesel and gasoline) is the source for Flt, Pyr, Chry, BPer, BaPyr, 1-NNap, 2-NNap, BaFlu-11, BaAnt-7,12, and BZT at the UNIS location.

A second group of compounds (Phe, Flu and 9,10-AntQ) was strongly correlated with the coal combustion ratio Flu/(Flu+Pyr) (Fig. 5, Table S28), supported by their predominance (along with 9-Flu) in the PP emissions (Fig. 2). 9-Flu may have other possible sources, including diesel and gasoline vehicle exhaust, coal powder, road dust particles (Keyte et al., 2013), and may be locally produced, transported from longer range, or secondarily formed in the atmosphere (Kojima et al., 2010). Interestingly, despite heavy rain during sampling, 9-Flu was found in its maximum concentration (about 2-fold higher the average detected level) on the second sampling day, which may suggest a strong local emission on that day in addition to the daily PP emissions. 9-Flu showed a strong positive correlation with Flt/(Flt+Pyr) ratio, indicating petrol or

marine fuel sources (Zhang et al., 2019). The ratio did not correlate with the traffic emitted compounds and the traffic ratio (group 1), thus marine fuel emission was considered as a potential source. Despite the intention to collect air samples at the end of summer to avoid the peak marine traffic, four large boats (fishing, two cruise vessels, and oil tanker; Fig. S5) and some private boats were registered in Longyearbyen harbor during sampling day 2 (Kystdatahuset, 2018). Our assumption is supported by reports of 9-Flu, cPPhen-4, and 9,10-AntQ as major oxy-PAHs in ship emissions (Czech et al., 2017; Zhao et al., 2020a; Zhao et al., 2019). According to Svalbard environmental law, vessels entering Svalbard coastal waters are required to use distillate marine fuel (DMA ISO 8217:2017) instead of heavy marine oil to satisfy regulations on the fuel sulfur content below 1.0 % (Governor of Svalbard, 2014). Ultra-low sulfur diesel CFPP-12 (NS-EN 590, with sulfur content below 0.001 %, Table S23) is also used for private boats. Such predominance of distillate marine diesel explains the strong correlation of 9-Flu with the marine fuel ratio and no correlation with 9,10-AntQ, which is mainly emitted from heavy fuel oil (Huang et al., 2018). Use of high quality fuels decrease the emissions of particles (Anderson et al., 2015). A reduction of up to 94 % particulate PAH emission was reported when burning low-sulfur fuel compared to heavy fuel oil (Huang et al., 2018; Gregoris et al., 2016; Kotchenruther, 2017; Czech et al., 2017). This explains the absence of particle-bound PAHs correlating with the marine ratio Flt/(Flt+Pyr).

The two PCs explain 74 % of the total variance of the UNIS samples. PP coal burning, traffic, and marine shipping emissions are determined as potential sources of PAHs and nitro- and oxy-PAHs.

**5 Conclusion**

Results provide insights into local sources of atmospheric PAHs and nitro- and oxy-PAHs in Svalbard. Source markers for the coal-burning PP in Longyearbyen were determined, and generally low emissions of PAHs confirmed an efficient exhaust cleaning system. However, PAHs are emitted daily from coal-burning and, due to a large volume of flue gas emissions, the PP remains an important local anthropogenic source of atmospheric contaminants. Overall, nitro- and oxy-PAH concentrations were the same order of magnitude as detected at other background Scandinavian and European air sampling stations, and PAHs were one order of magnitude higher than in Ny-Ålesund, Svalbard. The gas/particle partitioning of PAHs and nitro- and oxy-PAHs was dependent on air temperature and humidity, and mainly impacted semi-volatile compounds with three and four aromatic rings. Vehicle and marine traffic were other contributors to PAH emissions. The results also revealed secondary atmospheric formation as an additional source of 2+3-NFlt and 9-NAnt. The present study contributes to understanding fate and distribution of PAHs in the Arctic, and it provides important information on the phase-separated concentrations of PAHs and nitro- and oxy-PAHs in Arctic air, as well as markers of the Longyearbyen PP emissions. This data can eliminate uncertainties in model predictions that aim to assess the extent and impacts of Arctic atmospheric contaminants. Furthermore, the knowledge on local emissions level can be important in case of temperature inversion in the lower atmosphere when vertical dilution is limited and contaminants are trapped near the ground, which may be adverse to public health.

*Data availability*. The dataset used in this paper is included in the Supplement, and further information is available from the corresponding author tatiana.drotikova@unis.no.

*Supplement*. The supplement related to this article is available online at: https://www.atmos-chem-phys-discuss.net/acp-2020-142/acp-2020-142-supplement.pdf.

*Author contribution*. RK, AKH, and HR designed the campaign. TD conducted the field and lab works. TD with support from AA and RK optimized, validated and performed GC analysis and further quantification. TD processed and interpreted

PCA outcome. TD prepared the manuscript with contributions from all co-authors. TD, AA, and AKH prepared the Supplementary materials section.

*Competing interests*. The authors declare that there is no conflict of interest.

*Acknowledgements*. We gratefully acknowledge Longyearbyen Lokalstyre (Longyearbyen Community Council), personally Kim Rune Røkenes (a former leader of Energyverket), for the support on performing the PP exhaust sampling. We also thank Rasmus Bøckman (Lokalstyre, Energyverket) for providing information on the PP system operating parameters;
Morten Hogsnes and Kristin Lundgjerdingen (Applica Test & Certification AS) for sharing their knowledge on PP flue gas sampling; Siiri Wickström (UNIS) for helping with weather prediction; Marcos Porcires (UNIS) for on-site installation of weather station; Øyvind Mikkelsen (Norwegian University of Science and Technology) for teaching on PCA topic; Malte Jochmann (UNIS/Store Norske) for fruitful discussions on Svalbard coal quality. This research was financially supported by UNIS, NMBU, and the Svalbard Environmental Protection Fund (AtmoPart project).

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

**Table 1.** Concentrations of PAHs (G+P) and percentage in the particulate phase (%PM) in Longyearbyen power plant, UNIS, and Adventdalen. Data are average, minimum and maximum, n=6 for each location[*].

| Compound name | Abbreviated name | Power plant | | | UNIS | | | Adventdalen | | |
|---|---|---|---|---|---|---|---|---|---|---|
| | | Mean | Min-Max | Mean | Mean | Min-Max | Mean | Mean | Min-Max | Mean |
| | | ng m$^{-3}$ | ng m$^{-3}$ | %PM | pg m$^{-3}$ | pg m$^{-3}$ | %PM | pg m$^{-3}$ | pg m$^{-3}$ | %PM |
| Naphthalene | Nap | 51.82 | 32.74-59.82 | 7.4 | <MDL | <MDL | - | <MDL | <MDL | - |
| Acenaphtylene | Acy | 2.30 | 1.22-3.80 | 0.6 | 16.89 | 7.14-29.15 | 0.0 | 2.40 | 1.10-5.13 | 0.0 |
| Acenaphthene | Ace | 0.87 | 0.30-2.18 | 8.4 | 48.48 | 24.29-72.99 | 0.0 | 3.84 | 1.25-6.62 | 0.0 |
| Fluorene | Flu | 7.61 | 3.68-12.16 | 4.6 | 170.50 | 136.5-236.0 | 1.1 | 59.96 | 38.49-95.82 | 1.8 |
| Phenanthrene | Phe | 27.32 | 12.01-44.87 | 5.6 | 409.20 | 368.5-470.0 | 6.5 | 236.30 | 191.7-270.8 | 3.7 |
| Anthracene | Ant | 1.06 | 0.23-2.13 | 0.0 | 18.04 | 12.29-25.52 | 0.0 | 14.25 | 10.46-19.33 | 3.5 |
| Fluoranthnene | Flt | 6.99 | 1.43-12.49 | 3.9 | 28.45 | 24.14-36.06 | 40.5 | 19.14 | 10.76-38.54 | 23.2 |
| Pyrene | Pyr | 4.40 | 1.08-7.35 | 8.3 | 39.47 | 30.72-47.84 | 26.8 | 27.17 | 20.91-35.89 | 15.8 |
| Benzo(a)anthracene | BaAnt | 0.13 | 0.04-0.20 | 0.0 | 2.17 | 0.01-5.83 | 68.2 | n.d. | n.d. | - |
| Chrysene | Chry | 0.28 | 0.06-0.42 | 0.0 | 7.32 | 2.60-13.47 | 81.7 | 3.12 | 0.11-7.11 | 64.1 |
| Benzo(b+k)fluoranthene | BbkFlt | n.d. | n.d. | - | 2.23 | 0.01-5.87 | 100.0 | 0.75 | 0.01-2.79 | 100.0 |
| Benzo(a)pyrene | BaPyr | n.d. | n.d. | - | 0.89 | 0.01-2.46 | 100.0 | 0.34 | 0.01-1.16 | 100.0 |
| Indeno(1,2,3-cd)pyrene | IPyr | n.d. | n.d. | - | 1.63 | 0.07-3.79 | 100.0 | 0.71 | 0.07-2.67 | 100.0 |
| Dibenzo(a,h)anthracene | DBAnt | n.d. | n.d. | - | n.d. | n.d. | - | n.d. | n.d. | - |
| Benzo(g,h,i)perylene | BPer | n.d. | n.d. | - | 3.92 | 1.44-8.12 | 100.0 | 1.21 | 0.08-3.83 | 100.0 |
| ∑16 PAHs | | 102.8 | 61.9-139.1 | - | 749.2 | 687.4-866.9 | - | 369.1 | 279.0-454.5 | - |
| | | | | | | | | | | |
| 9-Fluorenone | 9-Flu | 12.35 | 5.57-19.54 | 19.2 | 270.30 | 128.2-543.8 | 41.7 | 139.40 | 110.2-177.2 | 25.5 |
| 9,10-Anthraquinone | 9,10-AntQ | 15.76 | 4.60-47.00 | 21.3 | 163.50 | 105.2-269.1 | 37.5 | 71.70 | 11.4-118.4 | 43.9 |
| 4H-Cyclopenta(def)-phenanthrene-4-one | cPPhe-4 | 1.30 | 0.51-2.55 | 15.8 | 27.23 | 20.16-35.80 | 65.5 | 18.77 | 11.97-39.10 | 38.1 |
| 9,10-Phenanthrenequinone | 9,10-PheQ | 2.13 | 0.96-4.40 | 0.0 | <MDL | <MDL | - | <MDL | <MDL | - |
| Benzo(a)fluoren-11-one | BaFlu-11 | 0.16 | 0.08-0.23 | 27.6 | 6.07 | 1.79-11.08 | 100.0 | 2.23 | 0.71-4.36 | 100.0 |
| Benzanthrone | BZT | 0.87 | 0.14-1.31 | 0.0 | 1.76 | 0.02-4.32 | 96.7 | 0.10 | 0.02-0.58 | 100.0 |
| Benzo(a)anthracene-7,12-dione | BaAnt-7,12 | n.d. | n.d. | - | 2.20 | 0.01-4.86 | 100.0 | 0.93 | 0.01-2.21 | 100.0 |
| 6H-Benzo(cd)pyren-6-one | BPyr-6 | n.d. | n.d. | - | n.d. | n.d. | - | n.d. | n.d. | - |
| ∑8 OPAHs | | 32.6 | 15.8-73.1 | - | 471.0 | 325.9-741.4 | - | 233.1 | 124.7-337.1 | - |
| | | | | | | | | | | |
| 1-Nitronaphthalene | 1-NNap | 2.19 | 0.99-4.69 | 61.7 | 16.97 | 13.36-21.53 | 0.1 | 5.02 | 1.91-9.84 | 1.5 |
| 2-Nitronaphthalene | 2-NNap | 0.26 | 0.11-0.40 | 31.7 | 5.08 | 2.44-7.33 | 3.1 | 1.88 | 1.29-2.83 | 5.4 |
| 2-Nitrobiphenyl | 2-NBip | 0.16 | 0.07-0.29 | 39.9 | 0.99 | 0.82-1.20 | 10.1 | 0.98 | 0.81-1.29 | 5.9 |
| 4-Nitrobiphenyl | 4-NBip | n.d. | n.d. | - | 2.23 | 1.51-2.68 | 0.0 | 2.45 | 0.29-4.10 | 0.0 |
| 1,5-Dinitronaphthalene | 1,5-DNNap | n.d. | n.d. | - | 0.80 | 0.05-2.17 | 80.0 | 0.93 | 0.05-3.72 | 53.9 |
| 5-Nitroacenaphthene | 5-NAce | n.d. | n.d. | - | 0.15 | 0.05-0.38 | 0.0 | 0.30 | 0.05-1.62 | 0.0 |
| 2-Nitrofluorene | 2-NFlu | 0.04 | 0.02-0.14 | 0.0 | 0.21 | 0.07-0.78 | 15.1 | 0.59 | 0.07-1.05 | 4.2 |
| 9-Nitroanthracene | 9-NAnt | 0.08 | 0.02-0.23 | 0.0 | 0.62 | 0.19-0.91 | n.d. | 2.26 | 0.12-4.70 | 57.8 |
| 9-Nitrophenanthrene | 9-NPhe | n.d. | n.d. | - | 0.20 | 0.09-0.37 | n.d. | 0.44 | 0.09-1.17 | 25.0 |
| 3-Nitrophenanthrene | 3-NPhe | 0.76 | 0.0003-1.93 | 96.1 | n.d. | n.d. | - | n.d. | n.d. | - |
| 2-Nitroanthracene | 2-NAnt | 0.31 | 0.07-0.62 | 0.0 | n.d. | n.d. | - | n.d. | n.d. | - |
| 2+3-Nitrofluoranthene | 2+3-NFlt | 0.52 | 0.06-1.14 | 0.0 | 9.50 | 7.32-11.37 | 94.5 | 12.30 | 4.68-26.66 | 79.8 |
| 4-Nitropyrene | 4-NPyr | 0.11 | 0.03-0.17 | 0.0 | n.d. | n.d. | - | n.d. | n.d. | - |
| 1-Nitropyrene | 1-NPyr | n.d. | n.d. | - | n.d. | n.d. | - | n.d. | n.d. | - |
| 2,7-Dinitrofluorene | 2,7-DNFlu | 0.06 | 0.001-0.14 | 0.0 | n.d. | n.d. | - | n.d. | n.d. | - |
| 7-Nitrobenzo(a)anthracene | 7-NBaAnt | 0.58 | 0.11-0.93 | 0.0 | n.d. | n.d. | - | n.d. | n.d. | - |
| 6-Nitrochrysene | 6-NChry | n.d. | n.d. | - | n.d. | n.d. | - | n.d. | n.d. | - |
| 1,3-Dinitropyrene | 1,3-DNPyr | n.d. | n.d. | - | n.d. | n.d. | - | n.d. | n.d. | - |
| 1,6-Dinitropyrene | 1,6-DNPyr | n.d. | n.d. | - | n.d. | n.d. | - | n.d. | n.d. | - |

| | | n.d. | n.d. | - | n.d. | n.d. | - | n.d. | n.d. | - |
|---|---|---|---|---|---|---|---|---|---|---|
| 1,8-Dinitropyrene | 1,8-DNPyr | n.d. | n.d. | - | n.d. | n.d. | - | n.d. | n.d. | - |
| 6-Nitrobenzo(a)pyrene | 6-NBaPyr | n.d. | n.d. | - | n.d. | n.d. | - | n.d. | n.d. | - |
| **∑22 NPAHs** | | 4.5 | 2.0-7.8 | - | 36.8 | 30.3-46.1 | - | 27.2 | 13.5-44.4 | - |

*Full results are given in SI (Table S9-S11)

<MDL below method detection limit

n.d. not detected

**Table 2.** Ratios of individual oxy- and nitro-PAHs to their corresponding parent PAHs (G+P) in Longyearbyen power plant; average of individual ratio values (n=6) with standard deviation are presented.

| Ratio | Mean±STD |
|---|---|
| **Nitro-PAH/PAH** | |
| 2-NFlu/Flu | 0.004±0.005 |
| 3-NPhe/Phe | 0.028±0.028 |
| 2-NAnt/Ant | 0.15±0.11 |
| 9-NAnt/Ant | 0.04±0.03 |
| 2+3-NFlt/Flt | 0.03±0.02 |
| 7-NBaAnt/BaAnt | 5.37±3.87 |
| | |
| **Oxy-PAH/PAH** | |
| 9,10-PheQ/Phe | 0.08±0.01 |
| cPPhen-4/Pyr | 0.31±0.11 |
| BaFlu-11/Chry | 0.65±0.34 |
| 9-Flu/Phe | 0.47±0.13 |
| 9-Flu/Flu | 1.67±0.29 |
| 9,10-AntQ/Ant | 12.17±7.31 |

**Table 3.** Source identification based on diagnostic ratios derived from total (G+P) concentrations; average of individual ratio values (n=6) with standard deviation are presented.

|  | Mean value | Potential source | Reference |
|---|---|---|---|
| **IPyr/(IPyr+BPer)** | | | |
| Power plant | n.d. | - | |
| UNIS | 0.32±0.01 | <0.35 Gasoline | (Ravindra et al., 2008b) |
| Adventdalen | 0.45±0.05 | 0.35-0.70 Diesel | (Kavouras et al., 2001), (Ravindra et al., 2008b) |
|  |  |  |  |
| **Flu/(Flu+Pyr)** | | | |
| Power plant | 0.64±0.11 | - | |
| UNIS | 0.81±0.04 | >0.5 Coal combustion | (Yunker et al., 2002), (Katsoyiannis and Breivik, 2014) |
|  |  | >0.5 Diesel | (Ravindra et al., 2008b) |
| Adventdalen | 0.68±0.05 | 0.64 Local power plant | This study |
|  |  | >0.5 Coal combustion | (Yunker et al., 2002), (Katsoyiannis and Breivik, 2014) |
|  |  |  |  |
| **Flt/(Flt+Pyr)** | | | |
| Power plant | n.d. | - | |
| UNIS | 0.42±0.02 | 0.31–0.42 Marine fuel | (Zhang et al., 2019) |
| Adventdalen | 0.40±0.08 | <0.5 Petrol emission | (Yunker et al., 2002) |
|  |  |  |  |
| **BbkFlt/BPer** | | | |
| Power plant | n.d. | - | |
| UNIS | 0.48±0.03 | <0.4 Gasoline | (Kuo et al., 2013) |
| Adventdalen | 0.87±0.25 | 0.78 Diesel | (Kuo et al., 2013) |

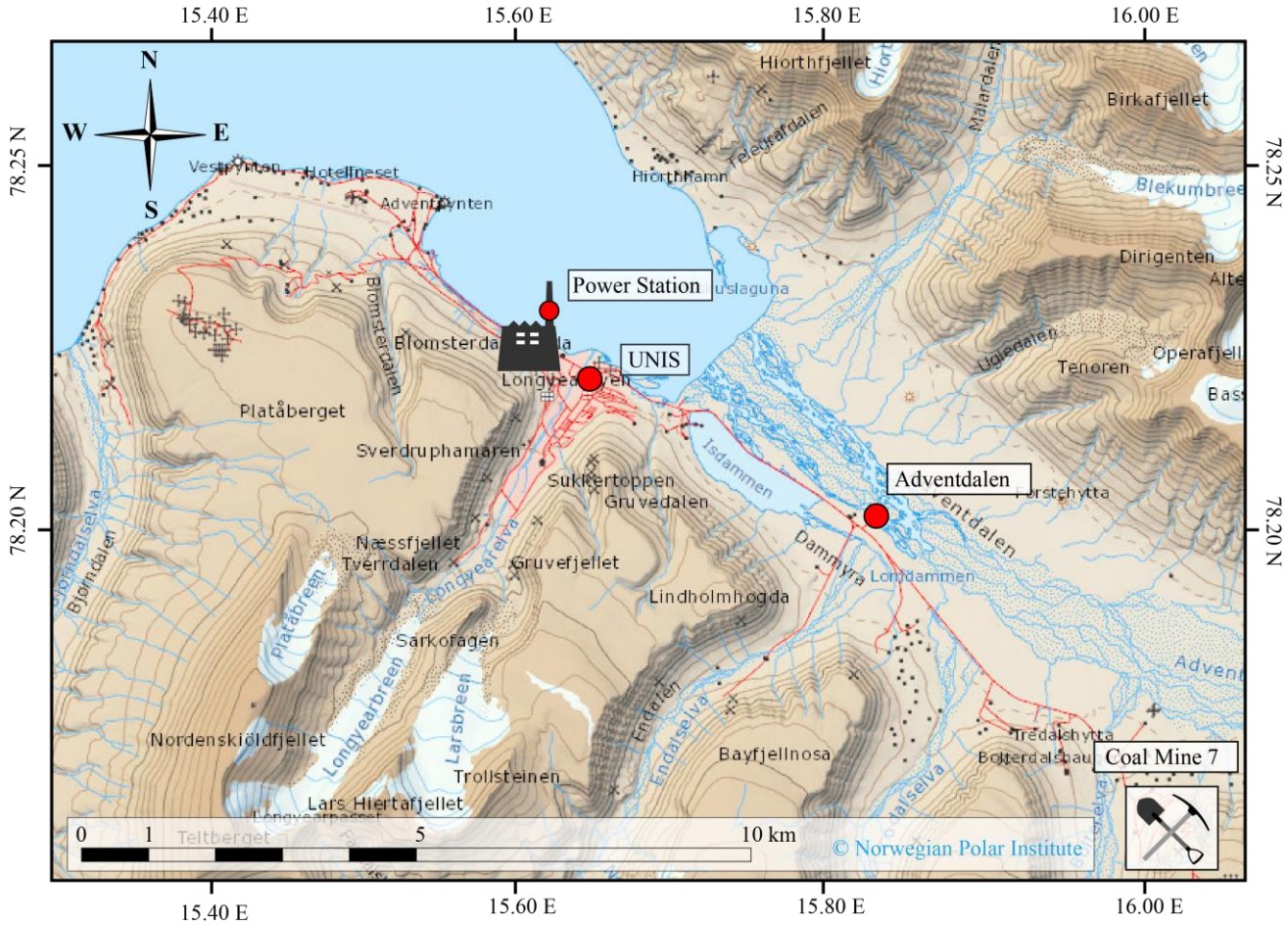

**Figure 1.** Air sampling transect locations in the vicinity of Longyearbyen.


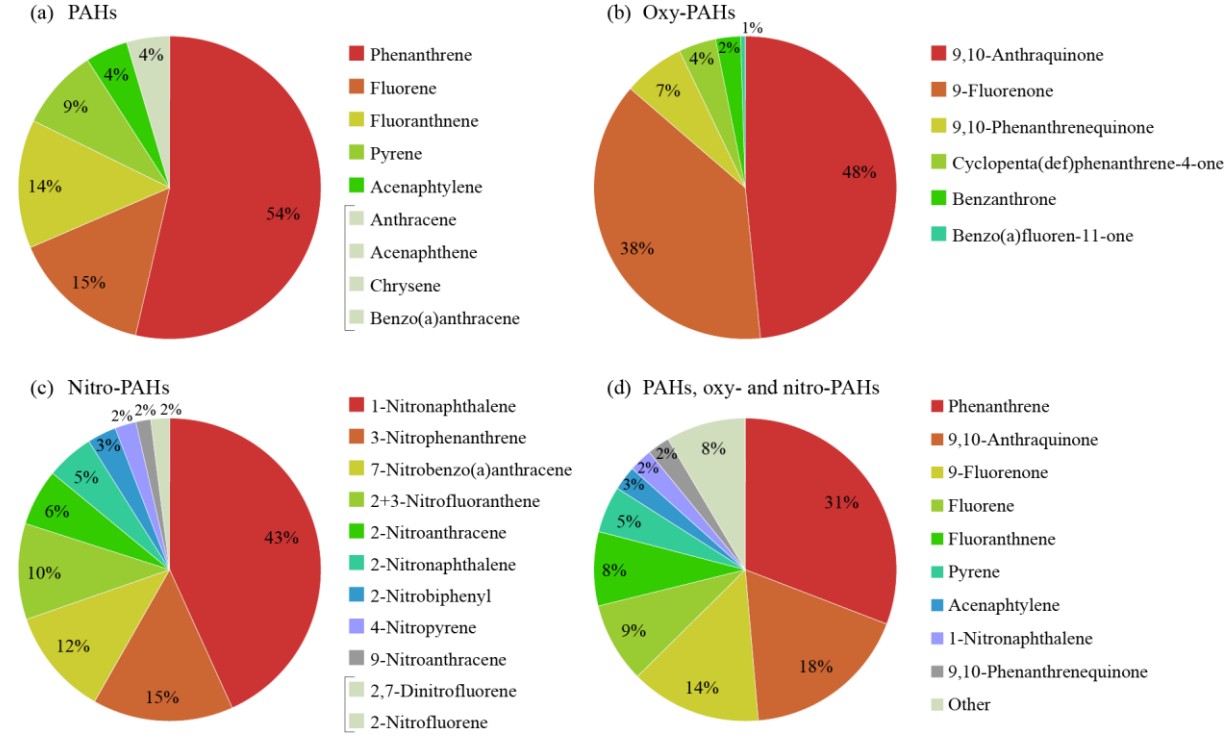

**Figure 2.** Proportion of PAHs and PAH derivatives (G+P; excluding Nap) in the Longyearbyen power plant emission (n=6).

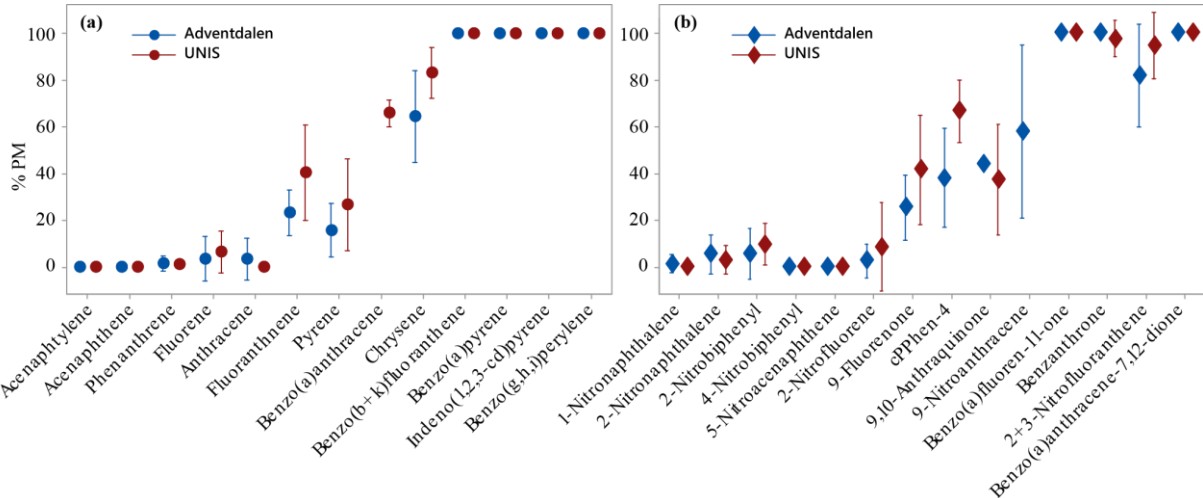

**Figure 3.** Percentage of (a) PAHs and (b) nitro- and oxy-PAHs determined in particulate phase (% PM) at UNIS (n=6) and Adventdalen (n=6); individual standard deviations are used to calculate the intervals.

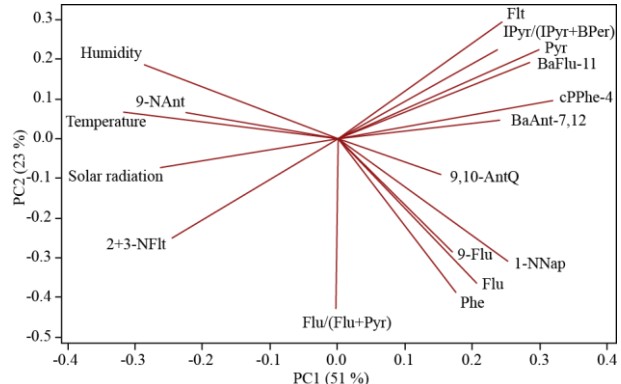

**Figure 4.** Principal component analysis loading plot of PC1 and PC2 for Adventdalen samples (G+P; n=6).


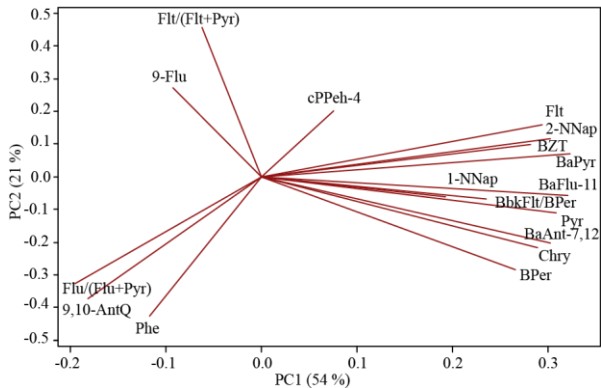

**Figure 5.** Principal component analysis loading plot of PC1 and PC2 for UNIS samples (G+P; n=6).