# Peer review of "Table of Contents"

_Atmospheric Chemistry and Physics, 2020_

## Referee Comment (RC1) · Anonymous Referee #1 · 29 Mar 2020

I enjoyed reading this paper; it is well written and scientifically sound. Furthermore, it makes an important contribution to the Arctic literature, providing results and interpretation of possible sources PAHs, oxy- and nitro-PAHs using ambient samples. I recommend the paper be accepted for publication with minor revisions.

Line 33 insinuates that the highest PAH levels observed in the Arctic are in the winter, and many of these would settle in the particulate phase because of the lower temperatures; in lines 55 it is suggested that the oxygenated and nitro PAHs would be higher in the summers due to the increased presence of oxidants and photodegradation. Wondering if there are actual environmental measurement studies that support this, or if

the authors have considered measuring ambient levels in this region in the summer.

Check phenanthrene spelling in introduction and wherever else relevant.

Figure 4: I think it's interesting that the PAHs that seem to exist mostly in the gaseous phase (e.g. PHE, FLU, Ant, 2NNap) tend to associate more with the lower temperature dry samples (A4 to A7). Curious as to whether the authors can clarify why.

Line 292-293: I can't seem to understand what the co-authors are trying to convey here. Why would A1 and A2 being negatively correlated indicate specific humidity is an essential parameter for the removal from the atmosphere? I thought it was interesting that A2 and A5 (the highest precipitation events) were quite separated on the PCA plot. Initially, I thought the differences could allude to the differences in scavenging efficiencies of snow and rain for PACs (i.e. where PAC concentrations in A5 lower than in A2?) But, then at the UNIS station you don't see the same separation for corresponding samples.

Could the captions of Figures 4-6 include what A1-A7 and U1-U7 represent? i.e. A1-A3 = humid, etc. It is stated in the text, but its easier for the readers to follow if these are added under the figures.

It would be interesting to include a table like Table S12 but a more comprehensive summary of PAHs levels in other Arctic regions. Because I found myself wondering how air concentrations were comparable to Arctic regions without a power-plant, for example.

---

## Referee Comment (RC2) · Anonymous Referee #2 · 2 Apr 2020

[referee-annotated manuscript omitted]

---

## Author Comment (AC1) · 6 Jun 2020

**Authors' Responses to Reviewers' Comments**

**Journal:** Atmospheric Chemistry and Physics
**Manuscript ID:** acp-2020-142
**Manuscript Title:** Polycyclic aromatic hydrocarbons (PAHs), oxy- and nitro-PAHs in ambient air of Arctic town Longyearbyen, Svalbard

**Authors:** Tatiana Drotikova, Aasim M. Ali, Anne Karine Halse, Helena C. Reinardy, Roland Kallenborn

We thank the two referees for their constructive comments, which have led to considerable improvement of the manuscript. After discussion within the author team, we have comprehensively addressed all recommendations provided. As it was suggested, multivariate statistical analysis was repeated with the PAH diagnostic ratios as variables. To focus on the initial goal of the section 4, discussions explaining the samples grouping and weather influences were omitted from the PCA outcomes. Therefore, we now introduce PCA loading plots instead of the biplots. General indications of the weather influence are now included in the "Gas/particle partitioning" section 3.2.2. We recently obtained new official information on the settlement's statistics of vehicles, fuel type, coal transportation, and marine port traffic, and this information is now included in the manuscript. Consequently, the "Source identification" section 4 was reorganized and rewritten in response to the comments from Reviewer 2, and we are convinced that now it provides better insights on the potential local sources of PAHs, nitro- and oxy-PAHs in Longyearbyen.

Below we provide our point-by-point responses (in blue color font) to each referee comment (in black color font; note "old" line numbers). Revised text is in red color font. Please, note new numeration of the revised text lines, as well as tables and figures in the supplement information to the manuscript. We hope that these details will clarify the current content and structure of the manuscript.

**Anonymous Referee #1**

Reviewer's general comment:

I enjoyed reading this paper; it is well written and scientifically sound. Furthermore, it makes an important contribution to the Arctic literature, providing results and interpretation of possible sources PAHs, oxy- and nitro-PAHs using ambient samples. I recommend the paper be accepted for publication with minor revisions.

Author's response:

We appreciate the reviewer's careful consideration of our manuscript and constructive comments. We have addressed the reviewer's specific comments below.

Reviewer's comment:

Line 33 insinuates that the highest PAH levels observed in the Arctic are in the winter, and many of these would settle in the particulate phase because of the lower temperatures in lines 55 it is suggested that the oxygenated and nitro PAHs would be higher in the summers due to the increased presence of oxidants and photodegradation. Wondering if there are actual environmental measurement studies that support this, or if the authors have considered measuring ambient levels in this region in the summer.

Author's response:

To the best of our knowledge, the data on oxy- and nitro-PAHs atmospheric formation in the high Arctic does not exist. Unfortunately, we can not provide any references on this topic. In currently planned follow-up studies we will report the year around obtained Svalbard data for PAHs, nitro- and oxy-PAHs, and we consider to include the atmospheric transformation of PAHs topic too.

Reviewer's comment:
Check phenanthrene spelling in introduction and wherever else relevant.
Author's response:
It was corrected twice in the abstract. Thank you.

Reviewer's comment on Figure 4:
I think it's interesting that the PAHs that seem to exist mostly in the gaseous phase (e.g. PHE, FLU, Ant, 2NNap) tend to associate more with the lower temperature dry samples (A4 to A7). Curious as to whether the authors can clarify why.
Author's response:
For LMW PAHs, water solubility (for Ace, Acy, Flu, Phe, 1- and 2-NNap, 9-Flu, and 9,10-AntQ) and polarity (for LMW nitro- and oxy-PAHs) play an additional role in wet scavenging processes (Shahpoury et al., 2018). The gas phase removal from the atmosphere is due to substance dissolution in water droplets (Shahpoury et al., 2018), which enhances the scavenging effect at higher humidity. The higher humidity conditions on the warmer days resulted in higher levels of LMW PAHs on the dry colder days. Although, as confirmed by earlier studies, snow scavenging may also be an important, and sometimes a dominating scavenging process for LMW PAHs (Wania et al., 1999). As can be seen on the plot provided below, concentrations of the LMW PAHs were reduced due to snowing event on the day 5. These observations show that this topic is quite complicated and requires long-term study investigation of LMW PAHs behavior in the high Arctic atmosphere. Wider consideration of the issue is outwith the scope of this study.

Another possible reason could be higher emissions from sources (cars, power plant) in colder period. Temporal variation in wind speed and wind direction might be of importance too but we did not reveal it in this study.

- *Shahpoury, P., Kitanovski, Z., Lammel, G. J. A. C., and Physics: Snow scavenging and phase partitioning of nitrated and oxygenated aromatic hydrocarbons in polluted and remote environments in central Europe and the European Arctic, Atmos. Chem. Phys., 18, 13495-13510, https://doi.org/10.5194/acp-18-13495-2018, 2018.*
- *Wania, F., Mackay, D., and Hoff, J. T.: The importance of snow scavenging of polychlorinated biphenyl and polycyclic aromatic hydrocarbon vapors, Environ. Sci. Technol., 33, 195-197, https://doi.org/10.1021/es980806n, 1999.*

[Figure]

Reviewer's comment on Adventdalen PCA:

I can't seem to understand what the co-authors are trying to convey here. Why would A1 and A2 being negatively correlated indicate specific humidity is an essential parameter for the removal from the atmosphere? I thought it was interesting that A2 and A5 (the highest precipitation events) were quite separated on the PCA plot. Initially, I thought the differences could allude to the differences in scavenging efficiencies of snow and rain for PACs (i.e. where PAC concentrations in A5 lower than in A2?) But, then at the UNIS station you don't see the same separation for corresponding samples.

Author's response:

Intensive precipitation occurred on day 2 (rain, 4.2 mm) and day 5 (snow, 2.8 mm) compared to the other days with little (A1 and A7, 0.1-0.2 mm) or no precipitation (A3 and A4). Hence, we expected that the samples A2 and A5, and perhaps A1 and A7 too, would be clustered all together due to scavenging effect and expected lower concentrations of PAHs and the derivatives on those days. Surprisingly, samples A2 and A5 were found to be separated and two groups of samples were observed. Within each group, A2 and A5 were clustered with other samples collected on the days with little or no precipitation. Therefore we concluded that the amount of precipitation influencing the PC1 to a lesser extent.

If considering the difference in scavenging efficiencies of snow and rain for PAHs as defining criteria, then the samples taken on the days without precipitation (A3 and A4) would expected to be separated from the others, and two (days with precipitation and days without precipitation) or three (rain, snow, no precipitation) groups would be observed. This PCA outcome was not seen.

[Figure]

Figure 1. PCA for Adventdalen samples

It was easy to notice that the samples taken on the cold days, later in September, were clustered together and separated from the warmer days. Arctic air is dry and cold, and ambient temperatures have strong influence on the actual amount of water vapor in the air (specific humidity). As consequence of similar ambient temperatures, specific humidity was found similar within each group (4.1±0.1 for the warm group and 2.8±0.7 for the cold group). Furthermore, all the studied compounds (except 2+3-NFlt, 2-NFlu and 9-NFlt) showed statistically significant ($p<0.05$) negative correlation with specific humidity (Table S17). In addition, the detected PAH concentrations were significantly lower on the rainy day 2 ($\sum$46 PAHs 430.9 pg m$^{-3}$) compared to the day 5 with snowing event ($\sum$46 PAHs 635.3 pg m$^{-3}$). The gas phase removal from the atmosphere is due to substance dissolution in water droplets, which enhances the scavenging effect at higher humidity. Hence, we suggest here that the actual amount of water vapor in the air might be an essential parameter for removal from the atmosphere, and attributed it to PC1.

However, this conclusion is only to be considered indicative based on the here preformed pilot study (with few observations only). Hence, further investigations are needed to confirm this assumption and add more statistical power to our hypothesis. Please note, ours study did not aim to focus on weather influence. We therefore have now removed all discussions explaining the samples grouping to focus on the potential sources identification. Thought brief indications of the weather influence are now included in the gas/particle partitioning section 3.2.2:

Page 7, line 281: The influence of wet deposition was indicated by a significant negative correlation between amount of precipitation and concentrations of several particle-bound HMW PAHs (Chry, BbkFlt, IPyr, BPer, BaFlu-11, and BaAnt-7,12) as well as semi-volatile Phe, Flt, and Pyr, which are more predominant in gaseous phase (Spearman correlation, $p<0.05$, Table S16. Effective wet scavenging of Phe, Flt, and Pyr has been suggested (Škrdlíková et al., 2011). Furthermore, a strong negative correlation with mass of water vapor in the air (specific humidity) was determined for most of the compounds (Spearman correlation, $p<0.05$, Table S17). Particle associated HMW compounds are readily scavenged by precipitation, while water solubility and polarity (for nitro- and oxy-PAHs) play an additional role in wet scavenging processes (Shahpoury et al., 2018). The gas phase removal from the atmosphere is due to substance dissolution in water droplets, which enhances the scavenging effect at higher humidity. Higher sensitivity of gas scavenging compared with particle scavenging towards liquid water content was also indicated by Škrdlíková et al. (2011).

Reviewer's comment on A2 and A5 samples separation:

But, then at the UNIS station you don't see the same separation for corresponding samples.

Author's response:

We did not expect to see similar PCA outcome for these two locations. The UNIS sampling station is located closer to the town and thus more influenced by emissions from passenger cars traffic, PP, marine port etc. Changes in the rate of emission from the sources due to human-influenced diurnal cycle may cause large variability in the PAH levels. Such disturbances are eliminated in Adventdalen, which is rural (an approximately background) sampling location with significantly less number of potential sources, and thus well suitable for such weather influences observations.

Reviewer's comment on the Figures 4-6:

Could the captions of Figures 4-6 include what A1-A7 and U1-U7 represent? i.e. A1- A3 = humid, etc. It is stated in the text, but its easier for the readers to follow if these are added under the figures.

Author's response:

As we explained, the discussion about weather influence was withdrawn from PCA. Now we introduced loading plots instead of the biplots. Please, see the updated version of the manuscript attached as separated file and the revised text is marked in red color font.

Reviewer's comment on the Table S12:

It would be interesting to include a table like Table S12 but a more comprehensive summary of PAHs levels in other Arctic regions. Because I found myself wondering how air concentrations were comparable to Arctic regions without a power-plant, for example.

Author's response:

Based on the comments from both the referees, we considered available studies on the Arctic as well as rural locations worldwide. The dedicated table with appropriate references is added for comparison. Please, see Table S14 in the updated version of the supplement information to the manuscript. We have added the following text in the Section 3.2.1:

Page 6, line 241: The measured PAH concentrations in the present study were in agreement with the 2 decades average data reported for the Arctic monitoring stations in Svalbard (Zeppelin) and Finland (Pallas), and were about an order of magnitude higher compared to the Canadian Arctic (Alert) concentrations (Yu et al., 2019). The PAH levels observed in Longyearbyen were significantly (up to 2 orders of magnitude) lower compared to rural sites in Europe and China (Table S14).

**Anonymous Referee #2**

Reviewer's comment:

This manuscript provides a study of PAHs and PAH derivatives (nitro- and oxy-PAHs) in the ambient air of two locations in Svalbard. The study includes results about their atmospheric concentrations, gas/particle partitioning and a tentative of source identification based on measurements in ambient air and at the emission of a coal power plant identified as a probable major contributor in the area of the study. Overall, the manuscript is well written, and the results are scientifically relevant especially in the context of the Arctic region which is still poorly documented about such toxic compounds. However, my major concerns are about the source identification methodology and results together with some statements about the gas/particle partitioning processes of these compounds. Thus, I would recommend the publication of this paper in ACP after some major revisions detailed directly into the pdf file of the text. Please also note the supplement to this comment: https://www.atmos-chem-phys-discuss.net/acp-2020-142/acp-2020-142-RC2- supplement.pdf.

Author's response:

We would like to thank the reviewer for his insightful comments towards improving our manuscript. We appreciate all the constructive suggestions and we have carefully addressed it, as explained below.

Lines 41-42: PAHs are regulated in many countries, eg. US, Canada, Holland, Sweden, Switzerland, and Denmark (Bandowe and Meusel, 2017).

Reviewer's comment:

Let's say USA, Europe and Canada. Specify in ambient air. Please cite the European Directive 2004/107/CE instead and for USA, USA EPA.

- *European Official Journal: Directive 2004/107/CE of the European Parliament and of the Council of 15 December 2004 relating to arsenic, cadmium, mercury, nickel and polycyclic aromatic hydrocarbons in ambient air (26/01/2005)., Official Journal, L23, 3–16, 2005.*
- *US EPA, 2011. Polycyclic Aromatic Hydrocarbons on the Gulf Coastline http://www.epa.gov/bpspill/pahs.html.*

Author's response:

We appreciate the comment and added information about PAH regulation guidelines. Text updated for ambient air regulation:

Page 1, line 40: Atmospheric PAHs are regulated in USA, Canada, UK, and Europe (EU Directive 2004/107/EC, 2005; US EPA, 2011; UK Air DEFRA, 2007; Ontario Ministry of the Environment and Climate Change, 2016).

Line 46: PAHs are byproducts of different incomplete combustion processes, mainly fossil fuels and biomass burning.

Reviewer's comment:

- *Ravindra, K., Sokhi, R. and Van Grieken, R.: Atmospheric polycyclic aromatic hydrocarbons: Source attribution, emission factors and regulation, Atmospheric Environment, 42, 2895–2921, doi: DOI:10.1016/j.atmosenv.2007.12.010, 2008.*

Author's response:

The reference was added. Thank you.

Line 56: PAHs react with a number of atmospheric oxidants, most notably the hydroxyl radical, ozone, the nitrate radical, and nitrogen dioxide (Keyte et al., 2013). This leads to their transformation into more toxic oxygenated and nitrated PAH derivatives (oxy-PAHs and nitro-PAHs). Oxy- and nitro-PAHs are also constituents of raw coal and can be emitted with PAHs following the same combustion processes (Huang et al., 2014b).

Reviewer's comment:

More generally from combustion processes including fossil fuel combustion, biomass burning, etc.

Author's response:

We mentioned in the text: "PAHs are byproducts of different incomplete combustion processes, mainly fossil fuels and biomass burning" and "Oxy- and nitro-PAHs are also constituents of raw coal and can be emitted with PAHs following the same combustion processes". These sentences are sufficiently explaining the potential oxy- and nitro-PAH sources from combustion processes. Thus, the text remained unchanged.

Line 61: the level of nitro- and oxy-PAHs in the Arctic atmosphere is unknown (Balmer and Muir, 2017).

Reviewer's comment:

is poorly documented

Author's response:

Based upon the reviewer's remarks, we have repeated the literature search regarding studies on atmospheric levels of nitro- and oxy-PAHs. Our updated literature survey confirmed that the statement of Balmer and Muir (2017) remains valid. Thus, the text was not changed.

Reviewer's comment, lines 99-103:

It would be useful for the reader to specify what are the typical emissions of the regulatory pollutants from this power plant (PM, SO2, NO2, etc.) especially with such treatment systems. It should refer to Table S12 for PM. This show the large efficiency of the flue gas treatments applied in this PP.

Author's response:

Additional information is provided in the main text. Text included in the manuscript line 103-104:

Page 3, line 103: Low emissions are reported: dust $1.5\pm0.2$, $SO_2$ $0.3\pm0.1$, $NO_x$ $244\pm19$, CO $63\pm5$ mg $Nm^{-3}$ (Lundgjerdingen, 2017), reflecting high efficiency of the flue gas cleaning system.

Reviewer's comment, line 112:

**WFGD** scrubber

Author's response:

Corrected accordingly

Reviewer's comment, line 115:

Tissuqaurtz? (no binding)

Author's response:

Yes, correct. Pallflex QFF, pure quartz, no binder, product ID 7202
https://shop.pall.com/us/en/products/zid7202?CategoryName=ID31&CatalogID=Laboratory. The updated text states:

Page 3, line 116: The particulate phase was collected on quartz fiber filter (QFF, pre-burned at 450°C for 6 h; Ø = 47 mm, no binder, Pallflex, USA), and the gaseous phase on polyurethane foam (PUF, Soxhlet pre-cleaned in toluene for 24 h followed by 24 h acetone wash; Ø = 50 mm, L = 75 mm, Klaus Ziemer GmbH, Germany).

Reviewer's comment, lines 116-117:
Using Soxhlet, PFE? Please provide some details
Author's response:
PUFs were Soxhlet pre-cleaned (VDI-cleaned, method BGI 505.47) by the producer, Klaus Ziemer GmbH, Germany. The cleaning method specification is now included in the text.

Reviewer's comment, line 118:
It is maybe not of great importance here. It should maybe discuss here. What is a typical PM size distribution from coal combustion especially after all the cleaning steps of the PP?
Author's response:
Mass-size distribution was not measured neither by the accredited company (Lundgjerdingen, 2017) working for PP nor by us. PM emission and PM size distribution depends on temperature of the ESP inlet flue gas (Li et al., 2018), voltage and current of ESP fields, number of electric fields, as well as retention time of ESP (Liu, 2018). Thus, the size distribution may vary largely, as well as PM emission itself shown in the Table S12. Removal efficiency is higher for larger particles, while finer particles are more difficult to eliminate (Li et al., 2018; Liu, 2018). According to the study by Li et al. (2018), coarse particles proportion in flue gas after ESP was reduced from 74 to 24%, while proportion of fine and ultra-fine (<1 μm) particles together increased from 10 to 38%, though medium size particles (2.5-10 μm) were the most abundant (about 40%) in the particulate emission. Besides, removal efficiency is higher by a combination of ESP and WFGD (up to 99.9% as stated in the manuscript lines 184-185) and the PM size distribution may be changed. Thus, we do not see how this literature based information (not measured by us) about PM size distribution can be incorporated into the PP sampling description section 2.2.1, as well as in the results discussion. We believe that the provided data about ultra-low particles emission (1.5 ± 0.2 mg m$^{-3}$; line 187) is sufficient for the recent study scope and, considering the provided PM emissions from other coal-fired power plants in Table S12, it also reflects well a great efficiency of the PP cleaning system.

Lines 122-123: The prevailing wind direction in Longyearbyen and Adventdalen is from the southeast. In summer, when the soil surface in Adventdalen becomes warmer than water surface in Adventfjørd, the wind direction can temporary change to northwesterly.
Reviewer's comment:
Any reference or data to support this statement?
Author's response:
Appropriate references added

- *Dekhtyareva, A., Edvardsen, K., Holmén, K., Hermansen, O., and Hansson, H. C.: Influence of local and regional air pollution on atmospheric measurements in Ny-Ålesund, International Journal of Sustainable Development and Planning, 11, 578-587, 10.2495/sdp-v11-n4-578-587, 2016.*
- *Esau, I., Argentini, S., Przybylak, R., Repina, I., and Sjöblom, A.: Svalbard meteorology, Advances in Meteorology, 2012, 8-164, 2012.*

Reviewer's comment, line 124:

the peak of marine

Author's response:

Corrected accordingly

Line 129: For each station, 6 high volume air samples were collected for particulate (QFF, pre-burned at 450°C for 6 h; Ø = 103 mm, no binder, Munktell/Ahlstrom, Finland)

Reviewer's comment:

Why not Pallflex like for the PP? No binder?

Author's response:

The providing company informed us that the Ø = 103 mm size Pallflex QFF for large volume air sampling would be out of stock for several month. Therefore, we simply chosen the similar type, no binder QFF available from a different company. The quality of the filters was similar: "MK 360 conditioned by high temperature pre-heating; Made of extremely pure micro-quartz fibers; Binder free; Particularly suitable for low amount of particles". The additional information "no binder" is now included in the text.

Lines 129-130: For each station, 6 high volume air samples were collected for particulate (QFF, pre-burned at 450°C for 6 h; Ø = 103 mm, Munktell/Ahlstrom, Finland) and gaseous (PUF, pre-cleaned in toluene for 24 h followed by 24 h acetone wash; Ø = 65 mm, L = 100 mm, Klaus Ziemer GmbH, Germany) phases.

Reviewer's comment:

Using Soxhlet, PFE? Please provide some details. Why not the same brand as for the PP?

Author's response:

The PUFs used for PP chimney and ambient air sampling are from the same producer (Klaus Ziemer GmbH, Germany) but different size because different pumps were used (portable low volume Digitel and High volume TISCH). The additional information on the cleaning method (Soxhlet) is now included in the text.

Reviewer's comment, line 135:

analysis

Author's response:

Corrected accordingly

Reviewer's comment, line 135:

(number of PP samples and blanks) to be specified in the previous section

Author's response:

To avoid repetition in Sections 2.2.1 and 2.2.2, extra details were included for clarification. Updated text:

Page 4, line 134: All samples (PP, UNIS, and Adventdalen) were kept intact inside the sampling unit after collection. In order to reduce the risk of post-collection contamination, the unit was sealed in two plastic bags for transportation to the lab, where samples were removed from the unit, sealed with layers of aluminum foil, and stored airtight in two plastic bags. Samples were kept frozen at -20°C until analysis. A total of 18 samples (18 QFFs and 18 PUFs) and 8 field blanks (4 for PP and 4 for UNIS and Adventdalen) were collected.

Reviewer's comment, line 138:

GC-EI-MS/MS and GC-ECNI-MS, give the names

Author's response:

The clarifying information was included. The updated text states:

Page 4, line 140: 16 PAHs, 8 oxy- and 21 nitro-PAHs (Table S3) were quantified using gas chromatography in combination with electron ionization mass spectrometry GC-EI-MS/MS and gas chromatography in combination with electron capture negative ion mass spectrometry GC-ECNI-MS, respectively.

Reviewer's comment, line 143:

QuEChERS-like extraction, full name?

Author's response:

The updated text includes this information:

Page 4, line 145: QFF samples were extracted with dichloromethane using a quick easy cheap effective rugged and safe QuEChERS-like procedure developed previously for the analysis of particulate bound PAHs (Albinet et al., 2013; Albinet et al., 2014).

Line 144: PUF samples were Soxhlet extracted with dichloromethane for 24 h.

Reviewer's comment:

Soxhlet extraction, manual or automatic Soxhlet?

Author's response:

All samples were manually Soxhlet extracted. The text remains the same.

Line 155-156: The method efficiency was tested using QFF (n=4) and PUF (n=4) spiked samples (Table S7). Acceptable recoveries ranged between 63-109% for dPAHs, 56-68% for dOxy-PAHs, and 44-89% for dNitro-PAHs (Table S8).

Reviewer's comment:

This is a first step but it would have better to evaluate the extraction efficiency by analyzing a standard reference material especially for the particulate phase

Author's response:

We agree, it might have been beneficial and will be the subject for follow-up studies. However, we do not intend to use our analytical approach for accredited reporting and thus consider the method as completely validated.

Reviewer's comment, line 175:

Actually, on Table 1, it would have been better to provide the emission factors for the power plant for all the individual species (PAHs and derivatives) instead of the concentration in the flue gas.

It would be also interesting to compare these results to results from the bibliography event for coal power plants with different flue gas treatment systems as done based on the concentrations below (Table S12)

Author's response:

The goal of analyzing the PP stack emission samples was only to obtain PAH chemical profile in order to determine the markers to be used for source identification. An estimation of the PP cleaning system efficiency would be a different study with different requirements (isokinetic sampling) and extra equipment for simultaneous measurements of PM, NOx, SOx emissions and perhaps some extra. We therefore decided to omit this interesting aspect for the present study to avoid overloading the manuscript.

For the determination of emission factors, as suggested, accurate data on the PP coal load during sampling is needed. Unfortunately, this internal information is not available to us. We can only operate with annual coal consumption but it varies between summer and winter seasons. Therefore, correct EFs could not be calculated, and also is considered as marginal for our study.

Lines 216-217: Nap is not a common indicator due to its ubiquitous presence and often high blank sampling material contamination; thus, Nap was not considered as potential marker.

Reviewer's comment:

Blank contamination by Nap, was observed in this study.

Author's response:

The text was corrected as follows:

Page 6, line 219: Due to its ubiquitous presence, Nap was not considered as suitable marker.

Line 218: Phe, Flu, Flt, Pyr, 9-Flu, and 9,10-AntQ were the main PAHs detected in the Longyearbyen PP flue gas

Reviewer's comment:

PACs. 9-Fluo and 9,10-Ant are oxy-PAHs

Author's response:

The corrected text states:

Page 6, line 219: Phe, Flu, Flt, Pyr, 9-Flu, and 9,10-AntQ were the main PAHs and oxy-PAHs detected in the Longyearbyen PP flue gas

Reviewer's comment on Figure 2, line 218:

I would probably modify this figure as follows. 4 pie charts: 1) proportions of PAHs, oxy-PAHs and nitro-PAHs over total PACs; 2) pie for PAHs only; 3) Pie for oxy-PAHs only; 4) Pie for nitro-PAHs only

Author's response:

The Figure 2 was modified as suggested

Line 226: The concentrations of PAHs measured at UNIS was a factor of 2 higher than at Adventdalen
Reviewer's comment:
to change "was" to "were"
Author's response:
Corrected accordingly

Line 226-227: The concentrations of PAHs measured at UNIS was a factor of 2 higher than at Adventdalen, while the PAH profiles were similar.
Reviewer's comment:
They (PAH chemical profiles) are not shown neither in the main text nor in the SM.
Author's response:
The PAH, oxy- and nitro-PAH chemical profiles are now included in the SI (Figure S2). The updated text states:
Page 6, line 232: The UNIS and Adventdalen chemical profiles of PAHs and oxy-PAHs were similar, while the profiles of nitro-PAHs differ (Fig. S2). Proportions of 1- and 2-NNap were higher in UNIS samples (about 60% of Σ22 nitro-PAHs), while 9-NAnt and 2+3-NFlt showed higher contributions into the nitro-PAH profile of samples from Adventdalen (about 55% of Σ22 nitro-PAHs).

Reviewer's comment, line 227:
the number of compounds quantified should be specified all along the text. Same for PP. Same for nitro-PAHs
Author's response:
The text is amended accordingly

Reviewer's comment, line 229:
the number of compounds quantified should be specified all along the text. Same for PP
Author's response:
The text is amended accordingly

Lines 231-233: The PAH concentrations measured at Longyearbyen (UNIS and Adventdalen) were two orders of magnitude higher than those detected at the Zeppelin station and the same order of magnitude as in Birkenes (southern mainland Norway) (NILU, 2019) (Table S13) for the same period (autumn 2018).
Reviewer's comment:
These results could be also compared to the ones reported by *Yu et al., 2019.*
- *Yu, Y., Katsoyiannis, A., Bohlin-Nizzetto, P., Brorström-Lundén, E., Ma, J., Zhao, Y., Wu, Z., Tych, W., Mindham, D., Sverko, E., Barresi, E., Dryfhout-Clark, H., Fellin, P. and Hung, H.: Polycyclic Aromatic Hydrocarbons Not Declining in Arctic Air Despite Global Emission Reduction, Environ. Sci. Technol., 53(5), 2375–2382, doi:10.1021/acs.est.8b*
Author's response:
A dedicated Table S14 with appropriate references is added for comparison. The text is adapted accordingly:

Page 6, line 240: The measured PAH concentrations in the present study were in agreement with the 2 decades average data reported for the Arctic monitoring stations in Svalbard (Zeppelin) and Finland (Pallas), and were about an order of magnitude higher compared to the Canadian Arctic (Alert) concentrations (Yu et al., 2019). The PAH levels observed in Longyearbyen were significantly (up to 2 orders of magnitude) lower compared to rural sites in Europe and China (Table S14).

Line 223: Table S13, UNIS and Adventdalen atmospheric concentrations (G+P) of ∑16 PAHs compared to national and regional background concentrations detected in autumn 2018.
Reviewer's comment:
In addition to this Table, it would be easier to show a graph presenting the average (+-SD) PAH chemical profiles for each site.
Author's response:
A graphic presentation is provided in the SI Figure S3 and the text added.
Page 6, line 239: The PAH profiles were dominated by Phe and Flu at all sites. A higher proportion of Phe was observed in Longyearbyen samples.

Lines 234- 235: Phe and Flu also dominated the PAH profile at Zeppelin station, which may indicate similar sources of contamination.
Reviewer's comment:
I won't be so conclusive in terms of sources
Author's response:
We agree, the revised text states:
Page 6, line 239: The PAH profiles were dominated by Phe and Flu at all sites. A higher proportion of Phe was observed in Longyearbyen samples.

Reviewer's comment, lines 236:
The data about nitro- and oxy-PAHs are scarce in the literature for the artic area but it would be interesting to compare these results in terms of concentrations; difference of order of magnitude between PAHs, oxy-PAHs, and nitro-PAHs; chemical profiles; with results from the literature for other locations worldwide (including for rural, altitude or remote sites). For instance, is the dominance of 9-Fluo and 9,10-AntQ is specific of your sites or it is something commonly observed in ambient air? Same for 1-NN and 2-NFlt? Same for both locations.
Author's response:
A dedicated Table S14 with appropriate references is added for comparison. The text is adopted accordingly. The revised text provides this comparison:
Page 7, line 249: The sum of oxy-PAHs detected at UNIS were similar to a rural site in eastern England (Alam et al., 2014), and a factor of 2 higher than at a rural location in the Czech Republic (Lammel et al., 2020), but were significantly lower than in rural southern China (Huang et al., 2014a) and the French Alps (Albinet et al., 2008).
Page 7, line 256: Overall, nitro-PAH concentrations were similar to those reported for the Pallas and Råö Scandinavian stations (Brorström-Lundén et al., 2010), and the rural site in the Czech Republic which is representative of central European background levels (Lammel et al., 2020).

Line 236-240: Concentrations of 9-Flu and 9,10-AntQ were the highest among measured oxy-PAHs in the present study. The 9-Flu level (270.3±146.9 pg m$^{-3}$ at UNIS and 139.4±24.9 pg m$^{-3}$ in Adventdalen) was the same order of magnitude as reported for the background monitoring stations in the north of Finland (Pallas) and in the south of Sweden (Råö) (Brorström-Lundén et al., 2010), while 9,10-AntQ (163.5±57.4 pg m$^{-3}$ at UNIS and 71.7±39.2 pg m$^{-3}$ in Adventdalen) was an order of magnitude higher in Longyearbyen. The nitro-PAH levels in our study were overall lower than other reported background sites.

Reviewer's comment on the last sentence:

That would need some references

Author's response:

The nitro-PAHs levels were compared with the same background stations data as for oxy-PAHs reported by Brorström-Lundén et al. (2010). The reference was already included. The revised text states:

Page 7, line 256: Overall, nitro-PAH concentrations were similar to those reported for the Pallas and Råö Scandinavian stations (Brorström-Lundén et al., 2010), and the rural site in the Czech Republic which is representative of central European background levels (Lammel et al., 2020).

Line 244: Gas/particle partitioning

Reviewer's comment:

As mentioned above, these results should be compared to the one available in the literature for other locations worldwide (in winter season to get comparable temperatures).

- *Delgado-Saborit, J. M., Alam, M. S., Godri Pollitt, K. J., Stark, C. and Harrison, R. M.: Analysis of atmospheric concentrations of quinones and polycyclic aromatic hydrocarbons in vapour and particulate phases, Atmospheric Environment, 77, 974–982, doi:10.1016/j.atmosenv.2013.05.080, 2013.*
- *Alam, M. S., Delgado-Saborit, J. M., Stark, C. and Harrison, R. M.: Using atmospheric measurements of PAH and quinone compounds at roadside and urban background sites to assess sources and reactivity, Atmospheric Environment, 77, 24–35, doi:10.1016/j.atmosenv.2013.04.068, 2013.*
- *Huang, B., Ming Liu, Bi, X., Chaemfa, C., Ren, Z., Wang, X., Sheng, G. and Fu, J.: Phase distribution, sources and risk assessment of PAHs, NPAHs and OPAHs in a rural site of Pearl River Delta region, China, Atmospheric Pollution Research, 5(2), 210–218, doi:10.5094/APR.2014.026, 2014.*
- *Albinet, A., Leoz-Garziandia, E., Budzinski, H., Villenave, E. and Jaffrezo, J.-L.: Nitrated and oxygenated derivatives of polycyclic aromatic hydrocarbons in the ambient air of two French alpine valleys: Part 1: Concentrations, sources and gas/particle partitioning, Atmospheric Environment, 42(1), 43–54, doi:10.1016/j.atmosenv.2007.10.009, 2008.*
- *Shahpoury, P., Lammel, G., Albinet, A., Sofuoglu, A., Dumanoglu, Y., Sofuoglu, S. C., Wagner, Z. and Zdimal, V.: Evaluation of a conceptual model for gas-particle partitioning of polycyclic aromatic hydrocarbons using poly-parameter linear free energy relationships, Environ. Sci. Technol., doi:10.1021/acs.est.6b02158, 2016.*
- *Tomaz, S., Shahpoury, P., Jaffrezo, J.-L., Lammel, G., Perraudin, E., Villenave, E. and Albinet, A.: One-year study of polycyclic aromatic compounds at an urban site in Grenoble (France): Seasonal*

*variations, gas/particle partitioning and cancer risk estimation, Science of The Total Environment, 565, 1071–1083, doi:10.1016/j.scitotenv.2016.05.137, 2016.*

Author's response:

We appreciate the constructive support of the reviewer and implemented the below reports in the discussion of the Gas/particle partitioning section. The additional text provided the comparison:

Page 8, line 291: In general, the obtained %PM were in agreement with the earlier reported (Table S18). Higher %PM of 9,10-AntQ and several nitro-PAHs (1- and 2-NNap, 2-NFlu, 9-NAnt, 9-NPhen) were detected in French alpine sites in winter (Albinet et al., 2008), while higher %PM of Flt, 9-Flu, and cPPhen-4 found in the present study contrasts with those reported for temperate urban and rural sites in China and Europe (Huang et al., 2014a; Tomaz et al., 2016). Sources difference, weather influence such as precipitation and temperature, as well as different atmospheric conditions (e.g., number of suspended particles, mass size particle distribution, and specific humidity), are likely responsible for these variations.

Lines 245-246: Gas/particle partitioning is an important process that controls transport, degradation, and distribution patterns of contaminants in and between environmental compartments (Huang et al., 2014).

Reviewer's comment:

This not the right reference to support this.

- *Franklin, J.; Atkinson, R.; Howard, P. H.; Orlando, J. J.; Seigneur, C.; Wallington, T. J.; Zetzsch, C., Quantitative determination of persistence in air. In Evaluation of Persistence and Long-Range Transport of Chemicals in the Environment; Klečka, G., et al., Ed.; SETAC Press: Pensacola, USA, 2000; pp 7-62.*
- *Bidleman, T. F. Atmospheric processes. Environ. Sci. Technol. 1988, 22 (4), 361-367.*
- *Finlayson-Pitts, B. J., Pitts, J.N. Chemistry of the Upper and Lower Atmosphere: Theory, Experiments, Application; Academic Press: San Diego, USA, 2000.*
- *Lammel, G.; Sehili, A. M.; Bond, T. C.; Feichter, J.; Grassl, H. Gas/particle partitioning and global distribution of polycyclic aromatic hydrocarbons – a modelling approach. Chemosphere 2009, 76 (1), 98-106.*

Author's response:

The reference list was updated. Thank you.

Lines 248-250: In general, LMW PAHs were found in the gas phase, while HMW PAHs were present in the particulate phase (Table 1), which is in accordance to their physico-chemical parameters, such as octanol-air partition coefficient (Table S3).

Reviewer's comment:

Other parameters are also to consider like the PM chemical composition, the vapor pressure (and so temperature and molecular weight).

- *Pankow, J. F. Review and comparative analysis of the theories on partitioning between the gas and aerosol particulate phases in the atmosphere. Atmos. Environ. 1987, 21 (11), 2275-2283.*

- *Lohmann, R.; Lammel, G. Adsorptive and absorptive contributions to the gas-particle partitioning of polycyclic aromatic hydrocarbons: state of knowledge and recommended parametrization for modeling. Environ. Sci. Technol. 2004, 38 (14), 3793-3803.*
- *Shahpoury, P., Lammel, G., Albinet, A., Sofuoglu, A., Dumanoglu, Y., Sofuoglu, S. C., Wagner, Z. and Zdimal, V.: Evaluation of a conceptual model for gas-particle partitioning of polycyclic aromatic hydrocarbons using poly-parameter linear free energy relationships, Environ. Sci. Technol., doi:10.1021/acs.est.6b02158, 2016.*
- *Tomaz, S., Shahpoury, P., Jaffrezo, J.-L., Lammel, G., Perraudin, E., Villenave, E. and Albinet, A.: One-year study of polycyclic aromatic compounds at an urban site in Grenoble (France): Seasonal variations, gas/particle partitioning and cancer risk estimation, Science of The Total Environment, 565, 1071–1083, doi:10.1016/j.scitotenv.2016.05.137, 2016.*

Author's response:

The list of the relevant physical-chemical properties was extended accordingly. Weather influence is discussed later in this section. We revised the text as follows:

Page 7, line 264: In general, LMW PAHs were found in the gas phase, while HMW PAHs were present in the particulate phase (Table 1), which is in accordance to their physico-chemical parameters, such as octanol-air partition coefficient, vapor pressure, and molecular (Table S3) (Tomaz et al., 2016; Shahpoury, 2016).

Page 7, line 270: %PM also depends on aerosol surface area, organic matter, and black carbon content (Lohmann and Lammel, 2004).

Line 254: Strong negative correlations (Spearman coefficient > 0.65) of percentage of PAH determined in particulate phase (%PM) with ambient temperature and specific humidity were determined for Chry, 1-NNap, 2-NNap, cPPhe-4, and 2+3-NFlt, although they were not statistically significant (Table S14).

Reviewer's comment:

These compounds, and most of the compounds shown in Table S14, are mainly associated to the gas phase. In my opinion, it is difficult to conclude anything about the link between humidity and gas/particle partitioning based in such conditions.

Author's response:

We agree with this note. The text has now been amended to reflect temperature influence on the partitioning of the earlier listed in this paragraph compounds, for which large migration between the phases was observed. Potential influence of humidity is discussed later in this section (lines 284-290). The text was updated:

Page 7, line 266: Repartitioning between phases (Fig. 3) mainly impacted semi-volatile compounds with three and four aromatic rings (Flt, Pyr, BaAnt, Chry; 2-NFlu, 9-Flu, cPPhe-4, 9,10-AntQ, 9-NAnt, and 2+3-NFlt) as a response to changing meteorological conditions (Hu et al., 2019). Strong negative correlations of percentage of PAH determined in particulate phase (%PM) with ambient temperature were confirmed for most of these compounds (Table S15. %PM also depends on aerosol surface area, organic matter, and black carbon content (Lohmann and Lammel, 2004).

Lines 265-268: The influence of wet deposition was indicated by a significant negative correlation between concentrations of several HMW PAHs (Pyr, Chry, BbkFlt, IPyr, BPer, BaFlu-11, and BaAnt-7,12) and precipitation (Spearman correlation, $p<0.05$, Table S15), resulting in a lower amount of particle-bound PAHs transported from the town, and thus lower %PM in Adventdalen.

Reviewer's comment:

What about particulate nitro-PAHs like 2+3-NFlt?

Author's response:

The negative correlation between concentrations and amount of precipitation was also found for 2+3-NFlt (80% PM) but it was not significant (Spearman correlation = -0,145, $p<0.05$), and thus not included in the Table S15.

Reviewer's comment:

Why Flt has not the same behavior like Pyr? The authors should consider the following article in their discussion (paper cited later in the article, line 348):

- *Škrdlíková, L., Landlová, L., Klánová, J. and Lammel, G.: Wet deposition and scavenging efficiency of gaseous and particulate phase polycyclic aromatic compounds at a central European suburban site, Atmospheric Environment, 45(25), 4305–4312, doi:10.1016/j.atmosenv.2011.04.072, 2011.*

Author's response:

We thank the reviewer for this insightful comment, and we think that these compounds do have similar behavior. We have now included compound specific aspects and discussions on environmental behavior of these compounds in the dedicated section. The new text states:

Page 7, line 281: The influence of wet deposition was indicated by a significant negative correlation between amount of precipitation and concentrations of several particle-bound HMW PAHs (Chry, BbkFlt, IPyr, BPer, BaFlu-11, and BaAnt-7,12) as well as semi-volatile Phe, Flt, and Pyr, which are more predominant in gaseous phase (Spearman correlation, $p<0.05$, Table S16. Effective wet scavenging of Phe, Flt, and Pyr has been suggested (Škrdlíková et al., 2011). Furthermore, a strong negative correlation with mass of water vapor in the air (specific humidity) was determined for most of the compounds (Spearman correlation, $p<0.05$, Table S17). Particle associated HMW compounds are readily scavenged by precipitation, while water solubility and polarity (for nitro- and oxy-PAHs) play an additional role in wet scavenging processes (Shahpoury et al., 2018). The gas phase removal from the atmosphere is due to substance dissolution in water droplets, which enhances the scavenging effect at higher humidity. Higher sensitivity of gas scavenging compared with particle scavenging towards liquid water content was also indicated by Škrdlíková et al. (2011).

Line 269: "Source identification" section

Reviewer's comment:

Reviewer's comment on "Source identification" section: For all of this part, I would suggest the authors instead of running the PCA only using the PAC concentrations, to also run it using the diagnostic ratios from Table 4 for both UNIS and Adventdalen. Separately or together with the concentrations (maybe not all the compounds but a selection of the key ones). Please also consider these publications in the choice of the ratios and in the discussion:

- *Dvorská, A., Lammel, G. and Klánová, J.: Use of diagnostic ratios for studying source apportionment and reactivity of ambient polycyclic aromatic hydrocarbons over Central Europe, Atmospheric Environment, 45(2), 420–427, doi:10.1016/j.atmosenv.2010.09.063, 2011.*
- *Katsoyiannis, A., Sweetman, A. J. and Jones, K. C.: PAH molecular diagnostic ratios applied to atmospheric sources: a critical evaluation using two decades of source inventory and air concentration data from the UK, Environ. Sci. Technol., 45(20), 8897–8906, doi:10.1021/es202277u, 2011.*

In addition, the authors should consider the ratios used in the study by Yu et al., 2019.

- *Yu, Y., Katsoyiannis, A., Bohlin-Nizzetto, P., Brorström-Lundén, E., Ma, J., Zhao, Y., Wu, Z., Tych, W., Mindham, D., Sverko, E., Barresi, E., Dryfhout-Clark, H., Fellin, P. and Hung, H.: Polycyclic Aromatic Hydrocarbons Not Declining in Arctic Air Despite Global Emission Reduction, Environ. Sci. Technol., 53(5), 2375–2382, doi:10.1021/acs.est.8b*

Author's response:

All the suggested papers were studied, and the ratios correlated best with the source specific PAHs were preferred. The applicability and scientific value of diagnostic ratios for PAH source elucidation was comprehensively discussed in the revised version. The revised text states:

Page 8, line 314: Diagnostic ratios may be affected by large-scale mixing of PAHs in the atmosphere, differing emission rates of PAH from the same source, influence of changing environmental conditions, and atmospheric processing of individual PAH compounds with different atmospheric lifetimes and reactivities (Alam et al., 2013; Tobiszewski and Namieśnik, 2012; Katsoyiannis and Breivik, 2014). Ratios based on highly reactive compounds such as Ant and BaAnt were not included, while more stable HMW PAHs diagnostic ratios were interpreted with greater confidence (Galarneau, 2008; Alam et al., 2014). Yunker et al. (2002) previously proposed the ratio of IPyr/(IPyr+BPer) to recognize vehicle from coal combustion emissions. BbkFlt/BPer was selected as an additional marker ratio for traffic due to the greater capacity to discriminate diesel and gasoline emissions, as well as its wider value range (Kuo et al., 2013). The Flt/(Flt+Pyr) ratio is widely used for source identification and, in particular, to understand if PAHs are mainly emitted from petroleum sources or from combustion processes (Yu et al., 2019). The Flu/(Flu+Pyr) ratio was selected as a specific indicator for coal combustion due to its strong correlation with the local PP determined markers, and the ratio value was also in agreement with literature (Yunker et al., 2002; Katsoyiannis and Breivik, 2014).

We are thankful for the suggestion to include the diagnostic ratios into PCA, which were adopted in the revised version. The multivariate statistical part was completely revised and rewritten (revised text is marked in red color font). Similarly to Albinet et al. (2007), we have removed the samples grouping interpretation because this discussion was distracting from the main point of our goal in this section. Our focus here is intended to be on the evaluation of the potential sources but not the weather influence. However, general discussion on the weather influences is now included in the gas/particle partitioning part (page 7, lines 281-292), as mentioned earlier.

- *Albinet, A., Leoz-Garziandia, E., Budzinski, H., Villenave, E., and Jaffrezo, J. L.: Nitrated and oxygenated derivatives of polycyclic aromatic hydrocarbons in the ambient air of two French alpine valleysPart 1: Concentrations, sources and gas/particle partitioning, Atmos. Environ., 42, 43-54, 10.1016/j.atmosenv.2007.10.009, 2008.*

Reviewer's comment, lines 278-279:

If the use of highly reactive compounds like BaA, BaP etc should be avoided, why the authors used the "diagnostic ratios" BaA/(BaA+Chry) and BaP/BPer or BaP/(BaP/Chry) (and in Table 3 Ant/(Ant+Phen)) in their source evaluation?

Author's response:

We were calculating those ratios because they have been applied and shown useful in previous studies. Hence, these calculations allow a better comparability with similar studies previously reported. However, these ratios are removed from the revised version of the manuscript.

Reviewer's comment, lines 281-282:

Just above it is mentioned: "Due to high atmospheric reactivity of Ant and BaAnt, utilization as source apportionment should be avoided". The authors should justify finally the use of this ratio. The authors should also consider the following publications about the use of the PAH diagnostic ratios in terms of source apportionment (and cite them to moderate their conclusions sometimes):

- *Dvorská, A., Lammel, G. and Klánová, J.: Use of diagnostic ratios for studying source apportionment and reactivity of ambient polycyclic aromatic hydrocarbons over Central Europe, Atmospheric Environment, 45(2), 420–427, doi:10.1016/j.atmosenv.2010.09.063, 2011.*
- *Katsoyiannis, A., Sweetman, A. J. and Jones, K. C.: PAH molecular diagnostic ratios applied to atmospheric sources: a critical evaluation using two decades of source inventory and air concentration data from the UK, Environ. Sci. Technol., 45(20), 8897–8906, doi:10.1021/es202277u, 2011.*

Author's response:

All the suggested papers were studied, and the ratios correlated best with the source specific PAHs were preferred. As example of the decision making: Dvorská et al. (2011) suggested Flt/(Flt+PYR) ratio as coal combustion characteristic but no correlation with the local PP detected marker was found (PCA was applied). Another suggested coal specific IPyr/(IPyr+BghiPer) ratio was not applicable for our study because IPyr and BghiPer were not detected in the local PP plume as a result of an effective exhaust cleaning system. Thus, other ratios were chosen.

Reviewer's comment, lines 290-291:

I have exactly the same comment as Reviewer 1. I would have expected different scavenging due to rainfalls or snowfalls. Were A5 concentrations lower than A2?

Author's response:

Yes, the reviewer assume correctly. The concentrations on the rainy day 2 were significantly lower compared to the detected levels on the day 5 with snow event, $\sum$46 PAHs 430.9 pg m$^{-3}$ on day 2 (rain) and ($\sum$46 PAHs 635.3 pg m$^{-3}$ on day 5 (snow). This was only true for Adventdalen. At UNIS the levels on the days 2 and 5 were nearly equal. The UNIS sampling station is located closer to the town and thus more influenced by emissions from different settlement sources. Higher emissions from any source on the day 2 can be a possible reason of non-observed scavenging effect of rain at UNIS. For instance, more boats were in the marine port of Longyearbyen during the day 2 sampling period (week 34 on the new

Figure S5) compared to all the other days. A passenger cruise ship Artania (MMSI: 311000608) was notably the biggest vessel (230 x 32 m, 44656 t gross tonnage) in the harbor on day 2. https://www.marinetraffic.com/en/ais/details/ships/shipid:371216/mmsi:311000608/imo:8201480/vessel:ARTANIA However, to focus on the potential sources only (the original goal of this section of the manuscript), the in-depth discussion on the difference between $2^{nd}$ and $5^{th}$ sampling days is now excluded since this topic is considered outside the scope of the study. More general discussion about weather influence is now included in the section 3.2.2 about the gas/particle partitioning.

Line 292: This indicates that the mass of water vapor in the air (specific humidity), in contrast to relative humidity, is an essential parameter for removal from the atmosphere.

Reviewer's comment:

Does it means like fog processing (by opposition to scavenging due to the rainfalls)? I suppose yes, so the following sentences (lines 299-303) should move here "The gas phase removal from the atmosphere is due to substance dissolution in water droplets (Shahpoury et al., 2018), which enhances the scavenging effect at higher humidity. A strong negative correlation with humidity was determined for all quantified LMW PAHs, significant for Acy, Pyr, 1-NNap, and 9-Flu (Spearman correlation, p<0.05; Table S17). Presence of Ant and Flt (gas-phase PAHs with low polarity and water solubility) in the group is likely due to the same source of origin."

Author's response:

We agree it is probably correct to equate elevated humidity with fog processing. However, as reported earlier, general discussion on this topic is now moved to the Section 3.2.2. The new text states:

Page 7, line 281: The influence of wet deposition was indicated by a significant negative correlation between amount of precipitation and concentrations of several particle-bound HMW PAHs (Chry, BbkFlt, IPyr, BPer, BaFlu-11, and BaAnt-7,12) as well as semi-volatile Phe, Flt, and Pyr, which are more predominant in gaseous phase (Spearman correlation, p<0.05, Table S16. Effective wet scavenging of Phe, Flt, and Pyr has been suggested (Škrdlíková et al., 2011). Furthermore, a strong negative correlation with mass of water vapor in the air (specific humidity) was determined for most of the compounds (Spearman correlation, p<0.05, Table S17). Particle associated HMW compounds are readily scavenged by precipitation, while water solubility and polarity (for nitro- and oxy-PAHs) play an additional role in wet scavenging processes (Shahpoury et al., 2018). The gas phase removal from the atmosphere is due to substance dissolution in water droplets, which enhances the scavenging effect at higher humidity. Higher sensitivity of gas scavenging compared with particle scavenging towards liquid water content was also indicated by Škrdlíková et al. (2011).

Reviewer's comment, line 304:

Let's not call that tracers cause it implies chemical stability, specificity to the emission sources that are not the case for any PAC. "based on some source indicators..."

Author's response:

This term is now removed as suggested.

Lines 304-305: Based on the tracers and their loadings (Table S16), PC1 can be assigned to local PP coal burning (Flu, Phe, Flt, Pyr, 9-Flu, 9,10-AntQ).

Is it based on the local PP chemical profile obtained here?  It should be specified

Author's response:

Yes, the most abundant compounds detected in the local PP emission were used as markers in this study. It is clarified in the revised text:

Page 8, line 323: The Flu/(Flu+Pyr) ratio was selected as a specific indicator for coal combustion due to its strong correlation with the local PP determined markers, and the ratio value was also in agreement with literature (Yunker et al., 2002; Katsoyiannis and Breivik, 2014).

Reviewer's comment, line 306:

What is the proportion of diesel vs gasoline vehicle engines in Svalbard?

Author's response:

Additional information is added as text and Table S19 in the updated SI.

Page 10, line 376: 1114 private cars were registered in Longyearbyen in 2018 (sbb.no), including old and modern (Euro 3-7 emission standard) technology cars, approximately equally balanced between gasoline and diesel fuel.

Reviewer's comment, lines 309-310:

This would need references to support the secondary origin of these compounds and their use in such case

Author's response:

Appropriate references are added.

Page 9, line 364: These results indicate atmospheric formation as an additional source of 9-NAnt and 2+3-NFlt, in agreement with other studies (Lin et al., 2015; Hayakawa et al., 2000; Shahpoury et al., 2018).

Line 321: Relative contribution of primary and secondary sources of nitro-PAHs could be tested by applying a 2-NFlt/1-NPyr ratio (Zielinska et al., 1989).

Reviewer's comment:

It is not the right reference. Prefer:

- *Ciccioli, P., Cecinato, A., Brancaleoni, E., Frattoni, M., Zacchei, P., Miguel, A.H., De Castro Vasconcellos, P., 1996. Formation and transport of 2-nitrofluoranthene and 2-nitropyrene of photochemical origin in the troposphere. J. Geophys.Res. 101, 19567e19581. http://dx.doi.org/10.1029/95JD02118.*
- *Albinet, A., Leoz-Garziandia, E., Budzinski, H., Villenave, E., 2007. Polycyclic aromatic hydrocarbons (PAHs), nitrated PAHs and oxygenated PAHs in ambient air of the Marseilles area (South of France): concentrations and sources. Sci. Total Environ.384, 280e292. http://dx.doi.org/10.1016/j.scitotenv.2007.04.028.*
- *Albinet, A., Leoz-Garziandia, E., Budzinski, H., Villenave, E., Jaffrezo, J.-L., 2008a. Nitrated and oxygenated derivatives of polycyclic aromatic hydrocarbons in the ambient air of two French*

*alpine valleys: Part 1: concentrations, sources and gas/particle partitioning. Atmos. Environ. 42, 43e54. http://dx.doi.org/10.1016/j. atmosenv.2007.10.009.*

- *Bamford, H.A., Baker, J.E., 2003. Nitro-polycyclic aromatic hydrocarbon concentrations and sources in urban and suburban atmospheres of the Mid-Atlantic region. Atmos. Environ. 37, 2077e2091. http://dx.doi.org/10.1016/S1352-2310(03) 00102-X.*
- *Ringuet, J., Albinet, A., Leoz-Garziandia, E., Budzinski, H., Villenave, E., 2012b. Diurnal/nocturnal concentrations and sources of particulate-bound PAHs, Oxy- PAHs and Nitro-PAHs at traffic and suburban sites in the region of Paris (France). Sci. Total Environ. 437, 297e305. http://dx.doi.org/10.1016/j.scitotenv. 2012.07.072.*
- *Tomaz, S., Jaffrezo, J.-L., Favez, O., Perraudin, E., Villenave, E. and Albinet, A.: Sources and atmospheric chemistry of oxy- and nitro-PAHs in the ambient air of Grenoble (France), Atmospheric Environment, 161, 144–154, doi:10.1016/j.atmosenv.2017.04.042, 2017.*

Author's response:

We thank the reviewer for the additional references and extended the citations accordingly

Reviewer's comment, lines 322-323:

Again, what is the proportion of diesel vs gasoline vehicle engines in Svalbard? Could you really specify diesel? If diesel is so predominant, 1-NPyr should have been detected. See the following paper:

- *Keyte, I. J., Albinet, A. and Harrison, R. M.: On-road traffic emissions of polycyclic aromatic hydrocarbons and their oxy- and nitro- derivative compounds measured in road tunnel environments, Science of The Total Environment, 566–567, 1131–1142, doi:10.1016/j.scitotenv.2016.05.152, 2016.*

Author's response:

The proportion of diesel and gasoline passenger cars is equal (Table S19) but, based on own experience as a local permanent resident of Longyearbyen, we confirm that passenger cars are rarely driven outside the settlement. Due to proximity to the active coal mine N7 (about 7 km from the Adventdalen sampling station, Figure 1), regular coal transportation (5-10 times per day) by trucks and occasional tourist busses are the main vehicles in Adventdalen, both diesel driven.

The suggested study by Keyte et al. (2016) is mainly on passenger vehicles and not well suitable for modern heavy duty vehicles, considering last five years advances in the exhaust aftertreatment including gas recirculation, diesel particulate filter, diesel oxidation catalyst, selective catalytic reduction, ammonia oxidation catalyst and others.

The mining company provided us with the specific documentation on fuel quality, engine technology type of the trucks used for coal transportation. We confirm that the vehicles are only 1-2 years old and operate on high quality diesel (new Table S23) used for all vehicles in Longyearbyen. Recent studies on a modern technology heavy duty vehicles report substantial reduction of nitro-PAH concentrations, and 1-NPyr is not the dominating compound.

For instance, Liu et al. (2015) reports 8-10 orders of magnitude mitigation of 6-NChry, 7-NBaAnt, 4-NPyr and other nitro-PAHs, and 2 orders of magnitude reduction of 1-NPyr level as result of the similar to Longyearbyen trucks exhaust aftertreatment system.

- *Liu, Z. G., Wall, J. C., Ottinger, N. A., and McGuffin, D.: Mitigation of PAH and Nitro-PAH Emissions from Nonroad Diesel Engines, Environ. Sci. Technol., 49, 3662-3671, 10.1021/es505434r, 2015.*

Thus, considering diesel motors modern technologies, 1,NPyr is not a strong marker of diesel emission from modern trucks, and due to actually low traffic in Adventdalen, 1-NPyr level was below the method detection limit. The new insights are now included in the text:

Page 9, lines 332-345: Because of the rural position, car traffic is much lower at this location. At the same time, due to the proximity to an active mine (Fig. 1), heavy-duty vehicles (coal trucks, tourist busses, geotechnical drilling machinery) are thus the main candidate source for PAH vehicle emissions. Produced coal is regularly delivered from the coal mine to PP and storage area in the harbor on a road situated in 150 m distance from the Adventdalen sampling station. Coal is transported by Volvo FH540 trucks (built in 2018-2020) driven on diesel CFPP-12 (NS-EN 590) (Nilssen P., Store Norske, personal communication; Table S23). The trucks have Euro 6 standard (HC 0.13, CO 1.5, $NO_x$ 0.4, PM 0.01 g (kWh)$^{-1}$) (DieselNet, 2020) compliant Volvo D13K engines (540 hp, 405 kW) engines fitted with exhaust gas recirculation, diesel particulate filter, diesel oxidation catalyst, selective catalytic reduction, and ammonia oxidation catalyst (Volvo Trucks, 2020). These allow high operation temperatures and high efficiencies in reducing particle and $NO_x$ emissions. Numerous studies showed substantial reduction in gaseous and particulate emissions of PAH, nitro- and oxy-PAHs as the result of such mitigation in particle and $NO_x$ emissions (Hu et al., 2013; Gerald Liu et al., 2010; Khalek et al., 2015; Huang et al., 2015). Up to 10 orders of magnitude reduction in emission from similar to Volvo D13K heavy-duty engine was reported for several nitro-PAHs (6-NChry, 1-NPyr, 2-NPyr, 4-NPyr, 7-NBaAnt) (Liu et al., 2015; Gerald Liu et al., 2010), which were not detected in the present study most likely due to low vehicle number in Adventdalen.

Lines 329-330: However, studies by Wania et al. (1999) report that snow scavenging may be an important, and sometimes a dominating scavenging process for lighter PAHs, mediated via a process of adsorption to the air-ice (Wania et al., 1999).

Reviewer's comment:

That should be specified before, lines 290-293.

Author's response:

The discussion on the difference between snow and rain scavenging efficiency is not included in the text anymore. Thus, this sentence was deleted.

Line 331: Traffic emission (mainly diesel exhaust) and the Longyearbyen coal-burning PP were the main local sources of PAHs and nitro- and oxy-PAHs in Adventdalen, and atmospheric transformation of PAHs is an additional source of nitro-PAHs.

Reviewer's comment:

(mainly diesel exhaust) to be moderated

Author's response:

We are convinced that the new provided above text supports this statement better than before. The sentence was not changed.

Lines 334-335: LMW PAHs were scavenged by snow, while the level of humidity was an essential parameter for total PAH removal from the atmosphere.

Reviewer's comment:

OK. See comment line 290 and the one from reviewer 1.

Author's response:

The discussion on the difference between snow and rain scavenging efficiency is not included in the text anymore. Higher sensitivity of gas scavenging than particle scavenging towards liquid water content was also concluded by *Škrdlíková, L., Landlová, L., Klánová, J., and Lammel, G.: Wet deposition and scavenging efficiency of gaseous and particulate phase polycyclic aromatic compounds at a central European suburban site, Atmos. Environ., 45, 4305-4312, https://doi.org/10.1016/j.atmosenv.2011.04.072, 2011.* Our proposal on the importance of humidity level on the total scavenging is in agreement with this long-term study.

Line 342: several of these compounds (BPer, IPyr, and BaFlu-11) were reported to be emitted after gasoline burning as well (Zielinska et al., 2004; Albinet et al., 2007)

Reviewer's comment to include the additional references:

- *Keyte, I. J., Albinet, A. and Harrison, R. M.: On-road traffic emissions of polycyclic aromatic hydrocarbons and their oxy- and nitro- derivative compounds measured in road tunnel environments, Science of The Total Environment, 566–567, 1131–1142, doi:10.1016/j.scitotenv.2016.05.152, 2016.*
- *Rogge,W.F., Hildemann, L.M., Mazurek,M.A., Cass, G.R., Simoneit, B.R.T., 1993a. Sources of fine organic aerosol 0.2. Noncatalyst and catalyst-equipped automobiles and heavy duty diesel trucks. Environ. Sci. Technol. 27, 636–651.*
- *Miguel, A.H., Kirchstetter, T.W., Harley, R.A., Hering, S.V., 1998. On-road emissions of particulate polycyclic aromatic hydrocarbons and black carbon from gasoline and diesel vehicles. Environ. Sci. Technol. 32, 450–455.*
- *Li, C.K., Kamens, R.M., 1993. The use of polycyclic aromatic hydrocarbons as source signatures in receptor modeling. Atmos. Environ. 27, 523–532.*

Author's response:

References added. Thank you.

Line 342: Absence of 1-NPyr and 2-NFlu, the principal compounds of diesel exhaust (Albinet et al., 2007), supported gasoline combustion as more dominant source as well.

Reviewer's comment:

It is the same at Adventdalen

Author's response:

For the UNIS sample location we now report general traffic emission (diesel and gasoline) as an important potential emission source. Although, explanations on the below detection limit level of 1-NPyr are provided:

Page 10, line 388: The diesel emission predominance was found for two out of the six sampling days, although particulate phase 1-NPyr, a marker of diesel emissions, was not detected. 1-NPyr forms in the

combustion chamber of diesel engines by the addition of nitrogen oxide or nitrogen dioxide to free Pyr radicals (IARC, 2014). Its generation is facilitated by the high engine temperatures (IARC, 2014; Karavalakis et al., 2010; Guan et al., 2017; Huang et al., 2015), which likely can not be reached in Longyearbyen due to short driving distances and low speed limit. The use of high quality ultra-low sulfur fuel with substantially reduced emissions of NOx leads to reduced nitration of PAHs during fuel combustion (Heeb et al., 2008; Zhao et al., 2020b), and together with low total vehicle number, resulting in low nitro-PAH emissions. Occurred atmospheric deposition may be of influence too.

Line 343: Absence of 1-NPyr and 2-NFlu, the principal compounds of diesel exhaust (Albinet et al., 2007), supported gasoline combustion as more dominant source as well.
Reviewer's comment to include additional reference:

- *Keyte, I. J., Albinet, A. and Harrison, R. M.: On-road traffic emissions of polycyclic aromatic hydrocarbons and their oxy- and nitro- derivative compounds measured in road tunnel environments, Science of The Total Environment, 566–567, 1131–1142, doi:10.1016/j.scitotenv.2016.05.152, 2016.*

Author's response:
1-NPyr and 2-NFlu are not used in this context in the revised version of the manuscript.

Lines 349-351: Zhang et al. (2015) reported that scavenging of particulate-phase PAHs is about 20 times more efficient than scavenging of gas phase PAHs.
Reviewer's comment:
This should be cited in the discussion above for Adventdalen (line 290-303)
Author's response:
This sentence does not exist in the updated version

Lines 353-354: The first group was characterized by the presence of 9,10-AntQ, Phe, and Flu, attributable to the PP emission source.
Reviewer's comment:
Is it based on the local PP chemical profile obtained here? It should be specified
Author's response:
Yes, it is based on the local PP emission profile. It was specified in the text.
Page 6, line 219: Further, Phe, Flu, Flt, Pyr, 9-Flu, and 9,10-AntQ were the main PAHs and oxy-PAHs detected in the Longyearbyen PP flue gas (Fig. 2), therefore the presence and diagnostic ratios (Table 3) of these compounds were used as markers of the PP source in the present work.

Line 354: Diagnostic ratios of Flu/(Flu+Pyr) and BaAnt/(BaAnt+Chry) (Table 4) also indicated that the PP is a source of PAHs and nitro- and oxy-PAHs at UNIS.
Reviewer's comment:
OK. Justify the use of such ratio BaAnt/(BaAnt+Chry) cause the sampling site is close to the PP and the time residence in the atmosphere to reach the sampling location is low.
Author's response:
The use of the BaAnt/(BaAnt+Chry) ratio was omitted as explained in the text (page 8, lines 314-326).

Reviewer's comment, line 355:

marker or indicator (not indicator)

Author's response:

The term is now changed all over the text.

Reviewer's comment, line 359:

I would be curious to see where the authors have seen that these compounds (4-NBip and 1,5-DNNap) are characteristics of diesel exhausts

Author's response:

Traffic emission source was concluded by (Alam et al., 2015), and heavy-duty diesel vehicles emission as source of 4-NBip by (Hu et al., 2013b). However, 4-NBip and 1,5-DNNap are not used in this context in the revised version.

- *Alam, M. S., Keyte, I. J., Yin, J., Stark, C., Jones, A. M., and Harrison, R. M.: Diurnal variability of polycyclic aromatic compound (PAC) concentrations: Relationship with meteorological conditions and inferred sources, Atmos. Environ., 122, 427-438, https://doi.org/10.1016/j.atmosenv.2015.09.050, 2015.*
- *Hu, S., Herner, J. D., Robertson, W., Kobayashi, R., Chang, M. C. O., Huang, S.-M., Zielinska, B., Kado, N., Collins, J. F., Rieger, P., Huai, T., and Ayala, A.: Emissions of polycyclic aromatic hydrocarbons (PAHs) and nitro-PAHs from heavy-duty diesel vehicles with DPF and SCR, Journal of the Air & Waste Management Association, 63, 984-996, 10.1080/10962247.2013.795202, 2013.*

Reviewer's comment, line 359:

This one is secondary (2+3-NFlt) and not at all from diesel exhausts

Author's response:

These two isomers, 2-NFlt and 3-NFlt, were not baseline separated on a chromatogram. Thus we report its sum concentration as 2+3-NFlt. According to Alam et al. (2015) 2-NFlt forms in atmosphere, while 3-NFlt was reported in several studies in diesel exhaust (Heeb et al., 2008; Liu et al., 2015), as well as several references are given in SI Table in (Keyte et al., 2013). That is why the presence of 2+3-NFlt in diesel emission was stated. However, 2+3-NFlt is not used in this context in the revised version.

- *Alam, M. S., Keyte, I. J., Yin, J., Stark, C., Jones, A. M., and Harrison, R. M.: Diurnal variability of polycyclic aromatic compound (PAC) concentrations: Relationship with meteorological conditions and inferred sources, Atmos. Environ., 122, 427-438, https://doi.org/10.1016/j.atmosenv.2015.09.050, 2015.*
- *Heeb, N. V., Schmid, P., Kohler, M., Gujer, E., Zennegg, M., Wenger, D., Wichser, A., Ulrich, A., Gfeller, U., Honegger, P., Zeyer, K., Emmenegger, L., Petermann, J.-L., Czerwinski, J., Mosimann, T., Kasper, M., and Mayer, A.: Secondary Effects of Catalytic Diesel Particulate Filters: Conversion of PAHs versus Formation of Nitro-PAHs, Environ. Sci. Technol., 42, 3773-3779, 10.1021/es7026949, 2008.*
- *Liu, Z. G., Wall, J. C., Ottinger, N. A., and McGuffin, D.: Mitigation of PAH and Nitro-PAH Emissions from Nonroad Diesel Engines, Environ. Sci. Technol., 49, 3662-3671, 10.1021/es505434r, 2015.*

- *Keyte, I. J., Harrison, R. M., and Lammel, G.: Chemical reactivity and long-range transport potential of polycyclic aromatic hydrocarbons – a review, Chem. Soc. Rev., 42, 9333-9391, https://doi.org/10.1039/C3CS60147A, 2013.*

Reviewer's comment, lines 360-362:

This would be more convincing based on the diagnostic ratio. I would suggest the authors instead of running the PCA only using the PAC concentrations, to also run it using the diagnostic ratios from Table 4 for both UNIS and Adventdalen.

Author's response:

The followed the suggestion and revised the PCAs of UNIS and Adventdalen data accordingly. It allowed better interpretation of the potential sources. We thank the reviewer for this valuable suggestion.

Lines 379-380: In contrast to the study by Yu et al. (2019), no strong indication for LRAT biomass burning emissions was found for this set of air samples.

Reviewer's comment:

For that, it would have been necessary to use other markers like retene for PAHs and even better, levoglucosan. Biomass burning from forest fires? It should be specified. Were any major forest fires during the sampling period? The authors should provide the back trajectories for the studied sampling period. The samplings were in summer so the impact of wood combustion for residential heating purposes from Northern Europe or Russia would be negligible.

Author's response:

This sentence was deleted. To note, we sampled on the days with predicted unusual NW wind (in order to catch the PP emission), thus air mainly arrived from the north and Greenland, the areas without biomass burning. We provided the back trajectories in the updated SI (Figure S4), and the updated text states:

Page 8, line 288: Due to changes in the Arctic front, more frequent precipitation, and very low levels of wood and coal burning for residential heating in the northern hemisphere in the summer, the LRAT of PAHs to the Arctic is low in summer. Sampling was performed on days with predicted northwesterly wind, and according to the 5-day back trajectory analysis, the air arriving to Svalbard in the sampling period mainly came from the north and from Greenland (Fig. S4). As discussed in Section 3.2.1, two orders of magnitude lower PAH concentrations were detected at the Zeppelin monitoring station compared to the levels in Longyearbyen on the same time. Thus, local emissions were the main sources of PAHs in Longyearbyen in this study.

Reviewer's comment on Table 4:

I would not use these ratios BaPyr/( BaPyr +Chry) and BaPyr/BPer cause of non-stability of BaPyr. In addition, they are not mentioned in the text.

Author's response:

This is corrected. We appreciate the constructive comment of the reviewer.

Reviewer's comment on Figure 2 title:

PAH and PAH derivatives.

Author's response:
Corrected accordingly.

Alam, M. S., Delgado-Saborit, J. M., Stark, C., and Harrison, R. M.: Using atmospheric measurements of PAH and quinone compounds at roadside and urban background sites to assess sources and reactivity, Atmos. Environ., 77, 24-35, https://doi.org/10.1016/j.atmosenv.2013.04.068, 2013.

Alam, M. S., Delgado-Saborit, J. M., Stark, C., and Harrison, R. M.: Investigating PAH relative reactivity using congener profiles, quinone measurements and back trajectories, Atmos. Chem. Phys., 14, 2467-2477, https://doi.org/10.5194/acp-14-2467-2014, 2014.

Alam, M. S., Keyte, I. J., Yin, J., Stark, C., Jones, A. M., and Harrison, R. M.: Diurnal variability of polycyclic aromatic compound (PAC) concentrations: Relationship with meteorological conditions and inferred sources, Atmos. Environ., 122, 427-438, https://doi.org/10.1016/j.atmosenv.2015.09.050, 2015.

Albinet, A., Leoz-Garziandia, E., Budzinski, H., and Viilenave, E.: Polycyclic aromatic hydrocarbons (PAHs), nitrated PAHs and oxygenated PAHs in ambient air of the Marseilles area (South of France): Concentrations and sources, Sci. Total Environ., 384, 280-292, https://doi.org/10.1016/j.scitotenv.2007.04.028, 2007.

Albinet, A., Leoz-Garziandia, E., Budzinski, H., Villenave, E., and Jaffrezo, J. L.: Nitrated and oxygenated derivatives of polycyclic aromatic hydrocarbons in the ambient air of two French alpine valleysPart 1: Concentrations, sources and gas/particle partitioning, Atmos. Environ., 42, 43-54, 10.1016/j.atmosenv.2007.10.009, 2008.

Albinet, A., Tomaz, S., and Lestremau, F.: A really quick easy cheap effective rugged and safe (QuEChERS) extraction procedure for the analysis of particle-bound PAHs in ambient air and emission samples, Sci. Total Environ., 450-451, 31-38, https://doi.org/10.1016/j.scitotenv.2013.01.068, 2013.

Albinet, A., Nalin, F., Tomaz, S., Beaumont, J., and Lestremau, F.: A simple QuEChERS-like extraction approach for molecular chemical characterization of organic aerosols: application to nitrated and oxygenated PAH derivatives (NPAH and OPAH) quantified by GC–NICIMS, Anal. Bioanal.Chem., 406, 3131-3148, https://doi.org/10.1007/s00216-014-7760-5, 2014.

Balmer, J., and Muir, D.: Polycyclic aromatic hydrocarbons (PAHs), in: AMAP Assessment 2016: Chemicals of emerging Arctic concern, edited by: Hung, H., Letcher, R., and Yu, Y., Arctic Monitoring and Assessment Programme (AMAP), Oslo, Norway, 219-238, 2017.

Borrás, E., Tortajada-Genaro, L. A., Vázquez, M., and Zielinska, B.: Polycyclic aromatic hydrocarbon exhaust emissions from different reformulated diesel fuels and engine operating conditions, 43, 5944-5952, 10.1016/j.atmosenv.2009.08.010, 2009.

Brorström-Lundén, E., Remberger, M., Kaj, L., Hansson, K., Palm Cousins, A., and Andersson, H.: Results from the Swedish national screening programme 2008, IVL Swedish Environmental Research Institute, Göteborg, Sweden, 69, 2010.

Dvorská, A., Lammel, G., and Klánová, J.: Use of diagnostic ratios for studying source apportionment and reactivity of ambient polycyclic aromatic hydrocarbons over Central Europe, Atmos. Environ., 45, 420-427, https://doi.org/10.1016/j.atmosenv.2010.09.063, 2011.

[revised manuscript text omitted]

NILU: Observation data of atmospheric PAHs at Zeppelin and Birkenes stations. Norwegian Institute for Air Research (Ed.), 2019.

Ontario Ministry of the Environment and Climate Change:  Ontario  Ambient  Air  Quality  Criteria. https://www.ontario.ca/page/ontarios-ambient-air-quality-criteria-sorted-contaminant-name, 2016.

Shahpoury, P., Kitanovski, Z., Lammel, G. J. A. C., and Physics: Snow scavenging and phase partitioning of nitrated and oxygenated aromatic hydrocarbons in polluted and remote environments in central Europe and the European Arctic, Atmos. Chem. Phys., 18, 13495-13510, https://doi.org/10.5194/acp-18-13495-2018, 2018.

Škrdlíková, L., Landlová, L., Klánová, J., and Lammel, G.: Wet deposition and scavenging efficiency of gaseous and particulate phase polycyclic aromatic compounds at a central European suburban site, Atmos. Environ., 45, 4305-4312, https://doi.org/10.1016/j.atmosenv.2011.04.072, 2011.

Tobiszewski, M., and Namieśnik, J.: PAH diagnostic ratios for the identification of pollution emission sources, Environ. Pollut., 162, 110-119, https://doi.org/10.1016/j.envpol.2011.10.025, 2012.

Tomaz, S., Shahpoury, P., Jaffrezo, J.-L., Lammel, G., Perraudin, E., Villenave, E., and Albinet, A.: One-year study of polycyclic aromatic compounds at an urban site in Grenoble (France): Seasonal variations, gas/particle partitioning and cancer risk estimation, Sci. Total Environ., 565, 1071-1083, 10.1016/j.scitotenv.2016.05.137, 2016.

UK Air DEFRA: The Air Quality Strategy for England, Scotland, Wales and Northern Ireland. Department for Environment, Food and Rural A. (Ed.), 2007.

US EPA: Polycyclic Aromatic Hydrocarbons on the Gulf Coastline. 2011.

Wania, F., Mackay, D., and Hoff, J. T.: The importance of snow scavenging of polychlorinated biphenyl and polycyclic aromatic hydrocarbon vapors, Environ. Sci. Technol., 33, 195-197, https://doi.org/10.1021/es980806n, 1999.

Westerholm, R., and Egebäck, K. E.: Exhaust emissions from light- and heavy-duty vehicles: chemical composition, impact of exhaust after treatment, and fuel parameters, 102, 13-23, 10.1289/ehp.94102s413, 1994.

Yu, Y., Katsoyiannis, A., Bohlin-Nizzetto, P., Brorström-Lundén, E., Ma, J., Zhao, Y., Wu, Z., Tych, W., Mindham, D., Sverko, E., Barresi, E., Dryfhout-Clark, H., Fellin, P., and Hung, H.: Polycyclic aromatic

hydrocarbons not declining in Arctic air despite global emission reduction, Environ. Sci. Technol., 53, 2375-2382, https://doi.org/10.1021/acs.est.8b05353, 2019.

Yunker, M. B., Macdonald, R. W., Vingarzan, R., Mitchell, R. H., Goyette, D., and Sylvestre, S.: PAHs in the Fraser River basin: a critical appraisal of PAH ratios as indicators of PAH source and composition, Org. Geochem., 33, 489-515, https://doi.org/10.1016/s0146-6380(02)00002-5, 2002.

Zhang, L., Cheng, I., Muir, D., and Charland, J. P.: Scavenging ratios of polycyclic aromatic compounds in rain and snow in the Athabasca oil sands region, Atmos. Chem. Phys., 15, 1421-1434, https://doi.org/10.5194/acp-15-1421-2015, 2015.